# Learning Rich Rankings

**Arjun Seshadri**
Stanford University
aseshadr@stanford.edu

**Stephen Ragain**
Stanford University
sragain17@gmail.com

**Johan Ugander**
Stanford University
jugander@stanford.edu

## Abstract

Although the foundations of ranking are well established, the ranking literature has primarily been focused on simple, unimodal models, e.g. the Mallows and Plackett-Luce models, that define distributions centered around a single total ordering. Explicit mixture models have provided some tools for modelling multimodal ranking data, though learning such models from data is often difficult. In this work, we contribute a *contextual repeated selection* (CRS) model that leverages recent advances in choice modeling to bring a natural multimodality and richness to the rankings space. We provide rigorous theoretical guarantees for maximum likelihood estimation under the model through structure-dependent tail risk and expected risk bounds. As a by-product, we also furnish the first tight bounds on the expected risk of maximum likelihood estimators for the multinomial logit (MNL) choice model and the Plackett-Luce (PL) ranking model, as well as the first tail risk bound on the PL ranking model. The CRS model significantly outperforms existing methods for modeling real world ranking data in a variety of settings, from racing to rank choice voting.

## 1 Introduction

Ranking data is one of the fundamental primitives of statistics, central to the study of recommender systems, search engines, social choice, as well as general data collection across machine learning. The combinatorial nature of ranking data comes with inherent computational and statistical challenges [15], and distributions over the space of rankings (the symmetric group $S_n$) are very high-dimensional objects that are quickly intractable to represent with complete generality. As a result, popular models of ranking data focus on parametric families of distributions in $S_n$, anchoring the computational and statistical burden of the model to the parameters.

Most popular models of rankings are distance-based or utility-based, where the Mallows [33] and Plackett-Luce [43] models are the two most popular models in each respective category. Both of these models simplistically assume transitivity and center a distribution around a single total ordering, assumptions that are limiting in practice. Intransitivities are frequent in sports competitions and other matchups [12]. The presence of political factions render unimodality an invalid assumption in ranked surveys and ranked voting, and recommender systems audiences often contain subpopulations with significant differences in preferences [26] that also induce multimodal ranking distributions.

A major open challenge in the ranking literature, then, has been to develop rich ranking models that go beyond these assumptions while still being efficiently learnable from data. Work on escaping unimodality is not new—the ranking literature has long considered models that violate unimodality (e.g., Babington Smith [50]), including explicit mixtures of unimodal models [39, 22]. However, such proposals are almost always restricted to theoretical discussions removed from practical applications.

In Figure 1 we provide a stylized visualization of multimodal data and models on the canonical Cayley graph of $S_5$ ($S_n$ with $n = 5$), contrasting a bimodal empirical distribution with the unimodal predicted probabilities from the Mallows and Plackett-Luce maximum likelihood estimates, as well

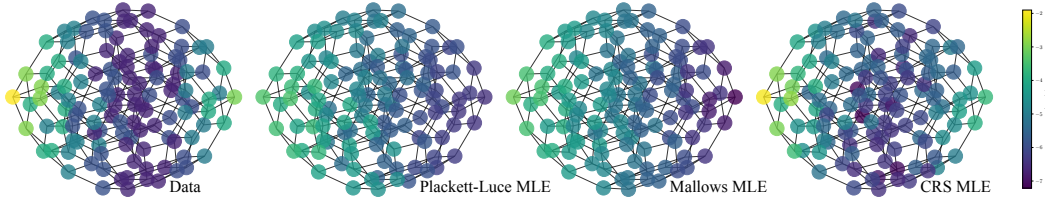

Figure 1: A synthetic multimodal distribution on the canonical Cayley graph of $S_5$ and the maximum likelihood estimates from the Plackett-Luce, Mallows, and full-rank CRS model classes.

the predicted probabilities of the model we introduce in this work, the *contextual repeated selection* (CRS) model.

An important tool for the modelling approach in this work is the transformations of rankings into choice data, where we can then employ tractable choice models to create choice-based models of ranking data. Building on the ranking literature on *L-decomposable distributions* [14], we conceptualize rankings as arising from a "top-down" sequence of choices, allowing us to create novel ranking models from recently introduced choice models. Both Plackett-Luce and Mallows models can be described as arising from such a top-down choice process [18]. We term this generic decomposition *repeated selection*. Estimating such ranking models reduces to estimating choice models on choice data implied by the ranking data, making model inference tractable whenever the underlying choice model inference is tractable.

Our contextual repeated selection (CRS) model arises from applying the recently introduced context-dependent utility model (CDM) [48] to choices arising from repeated selection. The CDM model is a modern recasting of a choice model due to Batsell and Polking [6], an embedding model of choice data similar to popular embedding approaches [42, 38, 47]. By decomposing a ranking into a series of repeated choices and applying the CDM, we obtain ranking models that are straightforward to estimate, with provable estimation guarantees inherited from the CDM.

Our theoretical analysis of the CRS ranking model builds on recent work giving structure-dependent finite-sample risk bounds for the maximum likelihood estimator of the MNL [49] and CDM [48] choice models. As a foundation for our eventual analysis of the CRS model, we improve and generalize several existing results for the MNL choice, CDM choice, and PL ranking models. Our work all but completes the theory of maximum likelihood estimation for the MNL and PL models, with expected risk and tail bounds that match known lower bounds. The tail bounds stem from a new Hanson-Wright-type tail inequality for random quadratic forms [25, 46, 27] with block structure (see Appendix, Lemma 3), itself of potential stand-alone interest. Our tight analysis of the PL tail and expected risk stems from a new careful spectral analysis of the (random) Plackett-Luce comparison Laplacian that arises when ranking data is viewed as choice data (see Appendix, Lemma 4).

Our empirical evaluations focus both on predicting out-of-sample rankings as well as predicting sequential entries of rankings as the top entries are revealed. We find that the flexible CRS model we introduce in this work achieves significantly higher out-of-sample likelihood, compared to the PL and Mallows models, across a wide range of applications including ranked choice voting from elections, sushi preferences, Nascar race results, and search engine results. By decomposing the performance to positions in a ranking, we find that while our new model performs similarly to PL on predicting the top entry of a ranking, our model is much better at predicting subsequent top entries. Our investigation demonstrates the broad efficacy of our approach across applications as well as dataset characteristics: these datasets differ greatly in size, number of alternatives, how many rankings each alternative appears in, and uniformity of the ranking length.

**Other related work.** There is an extensive body of work on modeling and learning distributions over the space of rankings, and we do not attempt a complete review here. Early multimodal ranking distributions include Thurstone's Case II model with correlated noise [51] from the 1920's and Babington Smith's model [50] from the 1950's, though both are intractable [35, 21]. Mixtures of unimodal models have been the most practical approach to multimodality to date [39, 22, 41, 3, 53, 13, 31], but are typically bogged down by expectation maximization (EM) or other difficulties.

Our approach of connecting rankings to choices is not new; repeated selection was first used to connect the MNL model of choice to the PL model of rankings [43]. Choice-based representations

of rankings in terms of pairwise choices are studied in *rank breaking* [5, 40, 28], whereas repeated selection can be thought of as a generalization,"choice breaking" *beyond* pairwise choices. The richness of the CRS model largely stems from the richness of the CDM choice model [48], one of several recent models to inject richness in discrete choice [45, 8, 7].

Our expected risk and risk tail bounds for maximum likelihood estimation stem from prior work for both the MLE for PL [24] and MNL [49] models. For MNL, risk bounds also exist for non-MLE estimators such as those based on rank breaking [4], LSR [37], and ASR [1]. However, all prior analyses (including for the MLE) fall short of *tight* guarantees (upper bounds that unconditionally match lower bounds). For the MNL model, Shah et al. [49] provides a tail bound for the pairwise setting and a (weak) expected risk bound for larger sets of a uniform size (that grows weaker for larger sets). Our results (tail and risk bounds) for MNL apply to any collection of variable-sized sets, a generalization that is itself necessary for our subsequent analysis of the PL and CRS models in a choice framework. Placing the focus back on rich ranking models, the tail and expected risk results for the CRS ranking model are the first of their kind for ranking models that are not unimodal in nature, meaningfully augmenting the scope of existing theoretical work on rankings.

## 2 Rankings from choices

We first introduce rankings, then choices, and develop the methodology connecting the two that is crucial to our paper's framework. Central to all three definitions is the notion of an item universe, $\mathcal{X}$, denoting a finite collection of $n$ items. Let $[n]$ denote the set of numbers $1, \ldots, n$, indexed by $i, j, k$.

**Rankings.** A ranking $\sigma$ orders the items in the universe $\mathcal{X}$, $\sigma : \mathcal{X} \mapsto [n]$. A ranking is also a bijection, letting us define $\sigma^{-1}(\cdot)$, the inverse mapping of $\sigma$. For any item $x \in \mathcal{X}$, $\sigma(x)$ denotes its rank, with a value of 1 indicating the highest position, and $n$ the lowest position. Similarly, the item in the $i$th rank is $\sigma^{-1}(i)$. A ranking distribution $P(\cdot)$ is a discrete probability distribution over the space of rankings $S_n$. That is, every ranking $\sigma \in S_n$ is assigned a probability, $0 \leq P(\sigma) \leq 1$, and $\sum_{\sigma \in \Sigma} P(\sigma) = 1$. A ranking model is then a particular representation of a ranking distribution $P$, parametric or not, including the Plackett-Luce, Mallows, and Thurstone models.

**Discrete choice.** Discrete choice modeling concerns itself with the *conditional* probability of a choice from a set $S \subseteq \mathcal{X}$, given that set $S$. That is, the modeling framework does not account for the process that $S$ is generated from (i.e., the probabilities different subsets may arise), but only the probability of choosing an item from a set, given that set a priori. Given a subset $S \subseteq \mathcal{X}$, a choice of $x \in S$ is denoted by the ordered pair $(x, S)$. The distribution of probabilities that $x$ is chosen from a given $S$ is denoted by $P(x|S)$, $\forall x \in S$, $\forall S \subseteq \mathcal{X}, |S| \geq 2$. That is, for every $S$, each $x \in S$ is assigned a probability $0 \leq P(x|S) \leq 1$, and $\sum_{x \in S} P(x|S) = 1$.

**Repeated selection.** Repeated selection follows a natural approach to constructing a ranking of the items of a set. Consider first the item that is preferred to all items and assign it rank 1. Then, of the items that remain, the item that is preferred is assigned rank 2. This assignment process is repeated until only one item remains, which is assigned rank $n$. In this way, a ranking is envisioned as a sequence of repeatedly identifying preferred items from a shrinking slate of options. When the sequence of choices are conditionally independent, we term this approach and its resulting interpretation *repeated selection*. Formally, a ranking distribution $P(\sigma)$ arising from repeated selection has the form

$$P(\sigma(x_1) = 1, \sigma(x_2) = 2, \ldots, \sigma(x_n) = n) = P(x_1|\mathcal{X})P(x_2|\mathcal{X} \setminus x_1) \cdots P(x_{n-1}|\{x_{n-1}, x_n\}).$$

It is easy to verify that any such distribution satisfies $\sum_{\sigma \in \Sigma} P(\sigma) = 1$. Under repeated selection, a ranking is converted into two objects of importance: a collection of choice sets, each a subset of the universe $\mathcal{X}$, as well as a sequence of independent choices conditioned on the choice sets. The latter (the conditional choice) is the subject of discrete choice modeling while the former (the collection) is a relatively unstudied random object that is a major focus of our analysis. The independence is worth emphasizing: the choices, conditioned on their choice sets, are treated as independent from one another. In contrast, the unconditioned choices are *not* independent from one another: certainly, knowledge of the first ranked item ensures that no other choice is that item.

Decomposing ranking distributions into independent repeated choices this way is not generic; see Critchlow et al. [14] for an extensive treatment of which ranking distributions can be *L-decomposed*

(decomposed from the "left"). As one example of its lack of generality, consider a process of *repeated elimination*, by which a choice model is applied as an elimination process, and the item to be first eliminated from a set is assigned the lowest rank, and the item to be eliminated from the set that remains, the second lowest, and so on. The resulting decomposition of the ranking (the "R-decomposition") generically induces an entirely different family of ranking distributions for a given family of choice models.

## 2.1 Popular examples of ranking via repeated selection

We illustrate two well known ranking models, and how they are a result of repeated selection applied to choice models. Both examples result in families of ranking distributions that center around a single total ordering—that is, the ranking distributions are unimodal.

**Plackett-Luce.** Perhaps the most popular discrete choice model is the Multinomial Logit (MNL) model, which describes the process of choice from a subset $S$ as simply a choice from the universe $\mathcal{X}$, conditioned on that choice being in the set $S$. This statement, along with some regularity conditions, is known as Luce's Choice Axiom [32]. That is,

$$P(x|S) = P(x|\mathcal{X}, x \in S) = \frac{P(x|\mathcal{X})}{\sum_{y \in S} P(y|\mathcal{X})} = \frac{\exp(\theta_x)}{\sum_{y \in S} \exp(\theta_y)},$$

where the final equality follows from setting $\theta_z = \log(P(z|\mathcal{X}))$, a popular parameterization of the model where $\theta \in \mathbb{R}^n$ are interpretable as utilities. By repeatedly selecting from the Multinomial Logit Model, we arrive at the Plackett-Luce model of rankings [43]:

$$P(\sigma(x_1) = 1, \sigma(x_2) = 2, \ldots, \sigma(x_n) = n \mid \theta) = \prod_{i=1}^{n-1} P(x_i|\mathcal{X} \setminus \cup_{j=1}^{i-1} x_j; \theta) = \prod_{i=1}^{n} \frac{\exp(\theta_{x_i})}{\sum_{j=i}^{n} \exp(\theta_{x_j})}.$$

The MNL model belongs to the broad class of independent Random Utility Models (RUMs) [34]. Any such RUM can be composed into a *utility-based ranking model* via repeated selection.

**Mallows.** The Mallows model assigns probabilities to rankings in a manner that decreases exponentially in the number of pairwise disagreements to a reference ranking $\sigma_0$. More precisely, under a Mallows model with concentration parameter $\theta$ and reference ranking $\sigma_0$, $P(\sigma; \sigma_0, \theta) \propto \exp(-\theta\tau(\sigma, \sigma_0))$, where $\tau(\cdot, \cdot)$ is Kendall's $\tau$ distance. The model can be fit into the framework of repeated selection via the choice model: $P(x|S) \propto \exp(-\theta|\{y \in S : \sigma_0(y) < \sigma_0(x)\}|)$ [18]. The model's reliance on a reference ranking $\sigma_0$ makes it generally NP-Hard to estimate from data [16, 9]. Mallows also belongs to a broader class of distance-based models, which replace Kendall's $\tau$ with other distance functions between rankings [14].

## 2.2 Beyond unimodality: contextual ranking with the CRS model

The recently introduced context-dependent utility model (CDM) of discrete choice [48] is both flexible and tractable, making it an attractive choice model to study in a repeated selection framework. The CDM models the probability of selecting an item $x$ from a set $S$ as proportional to a sum of pairwise interaction terms between $x$ and the other items $z \in S$. This strategy of incorporating a "pairwise dependence of alternatives" enables the CDM to subsume the MNL model class while also incorporating a range of context effects. Moreover, the matrix-like parameter structure of the CDM also opens the door for factorized representations that greatly improve the parametric efficiency of the model. The CDM choice probabilities, in full and factorized form, are then:

$$P(x|S) = \frac{\exp(\sum_{z \in S \setminus x} u_{xz})}{\sum_{y \in S} \exp(\sum_{z \in S \setminus y} u_{yz})} = \frac{\exp(\sum_{z \in S \setminus x} c_z^T t_x)}{\sum_{y \in S} \exp(\sum_{z \in S \setminus y} c_z^T t_y)},$$

where $u \in \mathbb{R}^{n(n-1)}$ represents the parameter space of the unfactorized CDM (a parameter for every ordered pair indexed by ordered pairs) and $T \in \mathbb{R}^{n \times r}$, $C \in \mathbb{R}^{n \times r}$ represents the parameter space of the factorized CDM, where $r$ is the dimension of the latent representations. Pushed through the repeated selection framework, we arrive at the CRS model of rankings, in full and factorized form:

$$P\big(\sigma(x_1) = 1, \ldots \sigma(x_n) = n\big) = \prod_{i=1}^{n} \frac{\exp(\sum_{k=i+1}^{n} u_{x_i x_k})}{\sum_{j=i}^{n} \exp(\sum_{k=j+1}^{n} u_{x_j x_k})} = \prod_{i=1}^{n} \frac{\exp(\sum_{k=i+1}^{n} c_{x_k}^T t_{x_i})}{\sum_{j=i}^{n} \exp(\sum_{k=j+1}^{n} c_{x_k}^T t_{x_j})}.$$

Just as the (factorized) CDM subsumes the MNL model (for every $r$), CRS subsumes the PL model. The benefits of a low-rank factorization are often immense in practice. The full CRS model can be useful, but its parameter requirements scale quadratically with the number of items $n$, and is therefore best applied only to settings where $n$ is small. The full CRS model is however conveniently amenable to many theoretical analyses, having a smooth and strongly convex likelihood whose landscape looks very similar to the Plackett-Luce likelihood. We thus focus our theoretical analysis of CRS on the full model, noting that all our guarantees that apply to the full CRS also apply to the factorized CRS. The factorized CRS model likely enjoys sharper guarantees for small $r$.

# 3 Better guarantees for MNL and Plackett-Luce

Efficient estimation is a major roadblock to employing rich ranking models in practice. This fact alone makes convergence guarantees—the type we provide in this section and the next—immensely valuable when assessing the viability of a model. Such guarantees for repeated selection ranking models involves both an analysis of the process by which a ranking is converted into conditionally independent choices, as well an analysis of the choice model that repeated selection is equipped with. While our efforts were originally focused on risk bounds for the new CRS model, in working to produce the best possible risk bounds for that model we identified several gaps in the analysis of more basic, widely used choice and ranking models. We first provide novel improved guarantees for existing foundational models, specifically, the MNL choice model and the PL ranking model, before proceeding to the CRS model in the next section. Relatively small modifications of the proofs in this section yield results for any utility-based ranking model (Section 2.1) that has a smooth and strongly convex likelihood.

In this section and the next, we focus on a ranking dataset $\mathcal{R} = \{\sigma_1, \ldots, \sigma_\ell\}$ of $\ell$ independent rankings each specified as a total ordering of the set $\mathcal{X}$ where $|\mathcal{X}| = n$. Given a repeated selection model of rankings generically parameterized by $\theta$, the likelihood for the dataset $\mathcal{R}$ becomes:

$$\mathcal{L}(\theta; \mathcal{R}) = \prod_{i=1}^{\ell} p(\sigma_i; \theta) = \prod_{i=1}^{\ell} \prod_{j=1}^{n} P(x_i | \mathcal{X} \setminus \cup_{k=1}^{j-1} x_k; \theta). \tag{1}$$

We can maximize the likelihood over $\theta$ to find the maximum likelihood estimate (MLE). Since the choices within each ranking are conditionally independent, the ranking likelihood reduces to a likelihood of a choice dataset with $\ell(n-1)$ choices. Finding the MLE of a repeated selection ranking model is thus equivalent to finding the MLE of a choice model. Because the PL and full CRS likelihoods are smooth and strongly log concave, we can efficiently find the MLEs in practice.

As a stepping stone to ranking, in Theorem 1 we first provide structure-dependent guarantees on the MLE for the underlying MNL choice models. Then, in Theorem 2 we analyze the set structure induced by repeated selection to provide guarantees on the PL ranking model of ranking data. This two-step process decouples the "choice randomness", or the randomness inherent to selecting the best item from the remaining set of items, from the "choice set randomness", the randomness inherent to the set of remaining items. All proofs are found in the Appendix.

**Multinomial logit.** The following theorem concerns the risk of the MLE for the MNL choice model, which evaluates the proximity of the estimator to the truth in Euclidean distance. We give both a tail bound and a bound on the expected risk.

**Theorem 1.** *Let $\theta^\star$ denote the true MNL model from which data is drawn. Let $\hat{\theta}_{MLE}$ denote the maximum likelihood solution. For any $\theta^\star \in \Theta_B = \{\theta \in \mathbb{R}^n : \|\theta\|_\infty \leq B, \mathbf{1}^T\theta = 0\}$, $t > 1$, and dataset $\mathcal{D}$ generated by the MNL model,*

$$\mathbb{P}\Big[ \Big\|\hat{\theta}_{MLE}(\mathcal{D}) - \theta^\star\Big\|_2^2 \geq c_{B,k_{max}} \frac{t}{m\lambda_2(L)^2}\Big] \leq e^{-t},$$

*where $k_{max}$ is the maximum choice set size in $\mathcal{D}$, $c_{B,k_{max}}$ is a constant that depends on $B$ and $k_{max}$, and $\lambda_2(L)$ depends on the spectrum of a Laplacian $L$ formed by $\mathcal{D}$. For the expected risk,*

$$\mathbb{E}\Big[ \Big\|\hat{\theta}_{MLE}(\mathcal{D}) - \theta^\star\Big\|_2^2\Big] \leq c'_{B,k_{max}} \frac{1}{m\lambda_2(L)^2},$$

*where $c'_{B,k_{max}}$ is again a constant that depends on $B$ and $k_{max}$.*

Focusing first on the expected risk bound, we see it tends to zero as the dataset size $m$ increases. The underlying set structure, represented in the bound by the object $\lambda_2(L)$, plays a significant role in the rate at which the bound vanishes. Here, $L$ is the Laplacian of the undirected weighted graph formed by the choice sets in $\mathcal{D}$. The algebraic connectivity of the graph, $\lambda_2(L)$, represents the extent to which there are good cuts in the comparison graph, i.e., whether all items are compared often to each other. Should there be more than one connected component in the graph, $\lambda_2(L)$ would be 0, and the bound would lose meaning. This behavior is not errant—the presence of more than a single connected component in $L$ implies that there is a non trivial partition of $\mathcal{X}$ such that no items in one partition have been compared to another, meaning that the relative ratio of the utilities could be arbitrarily large and the true parameters are *unidentifiable*.

The role of $\lambda_2(L)$ here is similar to Ford's [19] necessary and sufficient condition for MNL to be identifiable, that the directed comparison graph be strongly connected. The difference, however, is that $\lambda_2(L)$ depends only on the undirected comparison graph constructed only from the choice sets. The apparent gap between directed and undirected structure is filled by $B$, the bound on the true parameters in $\theta^\star$. As is natural, our bound also diverges if $B$ diverges. The remaining terms in the expression regulate the role of set sizes: larger set sizes increase algebraic connectivity, but make the likelihood less smooth, effects that ultimately cancel out for a balanced distribution of sets.

Theorem 1 is the first MNL risk bound that handles multiple set sizes, and is the first to be tight up to constants for set sizes that are not bounded by a constant. Our proof of the expected risk bound sharpens and generalizes the single-set-size proof of Shah et al. [49] to variable sized sets and largely follows the outline of the original proof, albeit with some new machinery (see e.g. Lemma 1, leveraging an old result due to Bunch–Nielsen–Sorensen [10], and the discussion of Lemma 1 in the proof of Theorem 1). A matching lower bound for the expected risk may be found in Shah et al., thus demonstrating the minimax optimality of the MLE at a great level of generality.

The tail bound component of the theorem is the first to go beyond pairwise comparisons. The result relies on a tail bound lemma, Lemma 3, that applies Hoeffding's inequality in ways that leverage special block structure innate to Laplacians built from choice data. This lemma replaces the use of a Hanson-Wright-type inequality in Shah et al.'s tail bounds for pairwise MNL. Lemma 3 leverages the fact that the constituent random variables are bounded, not merely subgaussian, to furnish a tail bound that is stronger than what Hanson-Wright-type tools deliver for this problem.

**Plackett-Luce.** With tight guarantees for the MLE of the MNL model, we proceed to analyze the PL ranking model. As Equation (1) demonstrates, the PL likelihood is simply a manifestation of the MNL likelihood. However, for rankings, the MNL tail bound provided so far is a random quantity, owing to the randomness of $\lambda_2(L)$. In choice, only the "choice randomness" is accounted for, and the choice sets are taken as given. In rankings, however, the choice sets themselves are random and we must therefore account for the "choice set randomness" that remains. We give expected risk bounds and tail bounds for the PL model in the following result.

**Theorem 2.** *Let $\mathcal{R} = \sigma_1, \ldots, \sigma_\ell \sim PL(\theta^\star)$ be a dataset of full rankings generated from a Plackett-Luce model with true parameter $\theta^\star \in \Theta_B = \{\theta \in \mathbb{R}^n : \|\theta\|_\infty \leq B, \mathbf{1}^T\theta = 0\}$ and let $\hat{\theta}_{MLE}$ denote the maximum likelihood solution. Assume that $\ell > 4\log(\sqrt{\alpha_B}n)/\alpha_B^2$ where $\alpha_B$ is a constant that only depends on $B$. Then for $t > 1$ and dataset $\mathcal{R}$ generated by the PL model,*

$$\mathbb{P}\left[\left\|\hat{\theta}_{MLE}(\mathcal{R}) - \theta^\star\right\|_2^2 \geq c_B'' \frac{n}{\ell}t\right] \leq e^{-t} + n^2\exp(-\ell\alpha_B^2)\exp\left(\frac{-t}{\alpha_B^2 n^2}\right),$$

*where $c_B''$ is a constant that depends on $B$. For the expected risk,*

$$\mathbb{E}\left[\left\|\hat{\theta}_{MLE}(\mathcal{R}) - \theta^\star\right\|_2^2\right] \leq c_B' \frac{n^3}{\ell}\mathbb{E}\left[\frac{1}{\lambda_2(L)^2}\right] \leq c_B \frac{n}{\ell},$$

*where $c_B' = 4\exp(4B)$, $c_B = 8\exp(4B)/\alpha_B^2$, and $L$ is the PL Laplacian constructed from $\mathcal{R}$.*

The expectation in the expected risk is taken over both the choices and choice set randomness, ensuring that the quantity on the final right hand side is deterministic. It is not difficult to show that $\lambda_2(L)$ is always positive (and thus $1/\lambda_2(L) < \infty$) for PL: every ranking contains a choice from the full universe, which is sufficient. Theorem 2 takes additional advantage of the fact that $B$ is often small, which results in subsets that are extraordinarily diverse, giving a considerably larger $\lambda_2(L)$ as

soon as the dataset has a sufficient number of rankings. The technical workhorse of Theorem 2 is Lemma 4, which provides a high probability lower bound on $\lambda_2(L)$ for the (random) Plackett-Luce Laplacian $L$.

Both our expected risk and tail bounds are the first bounds of their kind for the PL model, which matches a known lower bound on the expected risk (Theorem 1 in Hajek et al. [24]). Though the authors of that work claim to have bounds on expected risk that are weak by a $\log(n)$ factor, a closer inspection reveals that they only furnish upper bounds on a particular quantile of the risk. Much like our MNL tail bound, our PL tail bound integrates to a result on the expected risk that has the same parametric rates as our direct proof of the expected risk bound.

## 4 Convergence guarantees for the CRS model

The CRS model defines much richer distributions on $S_n$ than the PL model, but we are still able to demonstrate guaranteed convergence, a result that is the first of its kind for a non-simplistic model of ranking data. The focus of our study will be the full CRS model, statistical guarantees for which carry over to factorized CRS models of *any* rank.

Our analysis of the PL model required a generalized (to multiple set sizes) re-analysis of the MNL choice model. Similarly, we improve upon the known guarantees for the CDM choice model [48] that underlies the CRS ranking model by proving a tail bound in Lemma 7. Moreover, the added model complexity of the CDM creates new challenges, notably a notion of (random) "structure", in the structure-dependent bound, which does not simply reduce to analyzing a (random) Laplacian.

We first consider conditions that ensure the CRS model parameters are not underdetermined, conditions without which the risk can be arbitrarily large. Whereas the MNL model is immediately determined with choices from a single ranking—all the model requires is a single universe choice—a sufficient condition for CDM requires choices from all sets of at least 2 different sizes, with some technical exceptions (see [48], Theorem 1). Meeting this sufficient condition requires that at least $n$ rankings be present, since the two smallest collections of sets are the single set of size $n$ and the $n$ sets of size $n-1$. We demonstrate in Lemma 5 that, with high probability, $O(n\log(n)^2)$ rankings suffice to meet this sufficient condition. Of course, high probability does not mean always; and for the CRS model we more strongly rely on the assumption that the true parameter lies in a compact space to ensure that the risk is always bounded. Such assumptions are in fact always necessary for convergence guarantees of any kind, even for the basic MNL model [49].

We are now ready to present out main theoretical result for the CRS ranking model:

**Theorem 3.** *Let $\mathcal{R} = \sigma_1, \ldots, \sigma_\ell \sim CRS(u^\star)$ be a dataset of full rankings generated from the full CRS model with true parameter $u^\star \in \Theta_B = \{u \in \mathbb{R}^{n(n-1)} : u = [u_1^T, ..., u_n^T]^T; u_i \in \mathbb{R}^{n-1}, \|u_i\|_1 \leq B, \forall i; \boldsymbol{1}^T u = 0\}$ and let $\hat{u}_{MLE}$ denote the maximum likelihood solution. Assuming that $\ell > 8ne^{2B}\log(8ne^{2B})^2$, $c_B, ..., c_B'''$ are constants that depend only on $B$, and $t > 1$:*

$$\mathbb{P}\left[\|\hat{u}_{MLE}(\mathcal{R}) - u^\star\|_2^2 > \frac{c_B''' n^7}{\ell} t\right] \leq e^{-t} + n\exp\left(-t\min\left\{\frac{c_B'' n^6}{\ell}, 1\right\}\right)e^{-\ell/(8ne^{2B})}.$$

*For the expected risk,*

$$\mathbb{E}\left[\|\hat{u}_{MLE}(\mathcal{R}) - u^\star\|_2^2\right] \leq \mathbb{E}\left[\min\left\{\frac{c_B' n^3}{\ell\lambda_2(L)}, 4B^2 n\right\}\right] \leq c_B \frac{n^7}{\ell},$$

*where $L$ is a p.s.d. matrix constructed from $\mathcal{R}$.*

Similar to Theorem 2, the expectation is taken over both the choices and choice sets, rendering the final bound deterministic. The $L$ in the intermediate expression is not generally a graph Laplacian but rather a block structured matrix that captures the complex dependencies of the CDM parameters.

These expected and tail risk bounds may strike the reader as having a disappointing rate in $n$. Indeed, they leave us unsatisfied as authors. On one hand, modeling intransitivity, multimodality, and other richness comes at an inherent cost. The fact that any CRS model subsumes the PL model is also indicative of a slower rate of convergence. Despite these factors, in practice, as we demonstrate via simulations in Appendix C, the full CRS model appears to converge considerably faster, $O(n^2/\ell)$.

Table 1: Average out-of-sample negative log-likelihood for the MLE of repeated selection ranking models across different datasets (lowercase) or collections of datasets (uppercase), $\pm$ standard errors (of the mean) from five-fold cross-validation. Best results for each dataset appear in bold.

| | | | Ranking Model | | |
|---|---|---|---|---|---|
| Dataset | PL | CRS, $r = 1$ | CRS, $r = 4$ | CRS, $r = 8$ | Mallows (MGA) |
| sushi | $14.24 \pm 0.02$ | $13.94 \pm 0.02$ | $13.57 \pm 0.02$ | $\mathbf{13.47} \pm 0.02$ | $22.23 \pm 0.026$ |
| dub-n | $8.36 \pm 0.02$ | $8.18 \pm 0.02$ | $7.61 \pm 0.02$ | $\mathbf{7.59} \pm 0.02$ | $11.65 \pm 0.02$ |
| dub-w | $6.36 \pm 0.02$ | $6.27 \pm 0.02$ | $5.87 \pm 0.02$ | $\mathbf{5.86} \pm 0.01$ | $7.21 \pm 0.02$ |
| meath | $8.46 \pm 0.02$ | $8.23 \pm 0.02$ | $7.59 \pm 0.02$ | $\mathbf{7.56} \pm 0.02$ | $11.85 \pm 0.07$ |
| nascar | $113.0 \pm 1.4$ | $112.1 \pm 1.5$ | $103.9 \pm 1.8$ | $\mathbf{102.6} \pm 1.8$ | $238.5 \pm 0.3$ |
| LETOR | $12.2 \pm 1.0$ | $12.2 \pm 1.0$ | $10.5 \pm 1.1$ | $\mathbf{9.8} \pm 1.1$ | $22.5 \pm 0.5$ |
| PREF-SOC | $\mathbf{5.52} \pm 0.08$ | $5.53 \pm 0.07$ | $5.55 \pm 0.14$ | $5.54 \pm 0.15$ | $7.05 \pm 1.38$ |
| PREF-SOI | $4.1 \pm 0.1$ | $4.0 \pm 0.1$ | $\mathbf{3.9} \pm 0.1$ | $\mathbf{3.9} \pm 0.1$ | $6.8 \pm 0.2$ |

The factorized CRS model, used in our empirical work, appears to converge still faster, $O(nr/\ell)$. We believe the slow theoretical rates are likely a result of weakness in our analysis. The tightened analyses of the MNL choice and PL ranking models given in Theorem 1 and 2 are in fact by-products of trying to lower the bound in Theorem 3 as much as possible. The gap that still remains likely stems from a weak lower bound on the random "structure" of the CDM (Lemma 5).

The smoothness and strong convexity of the full CRS likelihood render it easy to maximize to obtain the MLE, making our result meaningful in practice. In contrast, MLE risk for ranking mixtures models is difficult to bound [41], and the separate difficulty of finding the MLE for mixtures [3] would question the value of such a result. Our bound on the expected risk extends to factorized CRS models, and despite the non-convexity of factorize likelihoods, gradient-based optimization often succeeds in finding global optima in practice and are widely conjectured to generally converge [20, 23, 30].

## 5 Empirical results

We evaluate the performance of various repeated selection models in learning from and making predictions on empirical datasets, a relative rarity in the theory-focused ranking literature. The datasets span a wide variety of human decision domains including ranked elections and food preferences, while also including (search) rankings made by algorithms. We find across all but one dataset that the novel CRS ranking model outperforms other models in out-of-sample prediction.

We study four widely studied datasets: the sushi dataset representing ranked food preferences, the dub-n, dub-w, and meath datasets representing ranked choice voting, the nascar dataset representing competitions, and the LETOR collection representing search engine rankings. We provide detailed descriptions of the datasets in Appendix A, as well as an explanation of the more complex PREF-SOC and PREF-SOI collections. Many of these datasets consist of top-$k$ rankings [17] of mixed length, which are fully amenable to decomposition through repeated selection.

### 5.1 Training

We use the stochastic gradient-based optimization method Adam [29] implemented in Pytorch to train our PL and CRS models. We run Adam with the default parameters ($lr = 0.001$, $\beta = (0.9, 0.999)$, $\epsilon = 1e - 8$). We use 10 epochs of optimization for the election datasets, where a single epoch converged. We cannot use Adam (or any simple gradient-based method), for the Mallows model as the reference permutation parameter $\sigma_0$ lives in a discrete space. Instead we select the reference permutation via the Mallows Greedy Approximation (MGA) as in [44], and then optimize the concentration parameter numerically, conditional on that reference permutation. Our results broadly show that the Mallows model, at least fit this way, performs poorly compared to all the other models, including even the uniform distribution (a naive baseline), so we exclude it from some of the more detailed evaluations.

For all datasets we use 5-fold cross validation for evaluating test metrics. Using the sushi dataset as an example, for each choice model we train on repeated selection choices for each of 5 folds of the

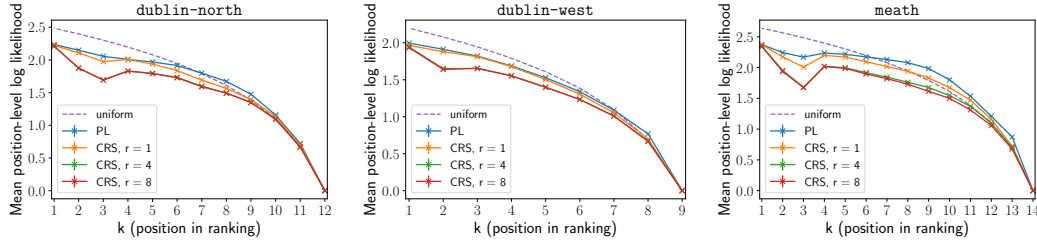

Figure 2: The mean position-level log likelihood of choice probabilities for the `dublin-north`, `dublin-west`, and `meath` election datasets.

5,000 rankings in the dataset. The optimization can be easily guided to exploit sparsity, parallelization, and batching. All replication code is publicly available[1].

## 5.2 Cumulative findings

In Table 1 we report average out-of-sample negative log-likelihoods for all datasets and collections, averaged over 5 folds. We see that across a range of dimensions $r$ the factorized CRS model typically offers significantly improved performance, or at least no worse performance, than the Mallows and Plackett-Luce models (where the CRS model generalizes the latter). For all datasets, the Mallows Greedy Approximation (MGA)-based model is markedly worse than the other models.

## 5.3 Position-level analysis

We next provide a deeper, position-level analysis of model performance. We measure the error at the $k$th position of a ranking $\sigma$ given the set of already ranked items by adding up some distance between the model choice probabilities for the corresponding choice sets and the empirical distribution of those choices in the data. For repeated selection models, we define the *position-level log-likelihood* at each position $k$ as $\ell(k, \theta; \sigma) := \log p_\theta(\sigma^{-1}(k), \{\sigma^{-1}(j)\}_{j \geq k})$. When averaging $\ell$ over a test set $T$ we obtain the mean position-level log-likelihood:

$$\ell(k; \theta, T) := \frac{1}{|T|} \sum_{\sigma \in T : len(\sigma) \geq k} \ell(k, \theta; \sigma), \tag{2}$$

where $len(\sigma)$ is $n$ for a full ranking and $k$ for a top-$k$ ranking.

In Figure 2 we analyze the election datasets at the position level, where we find that the CRS model ($r = 8$) makes significant gains relative to Plackett-Luce when predicting candidates near—but not at—the top of the list. We further notice that the performance is not monotonically decreasing in the number of remaining choices. Specifically, it is easier to guess the third-ranked candidate than the fourth, despite having fewer options in the latter scenario. A plausible explanation is that many voters rank candidates from a single political party and then stop ranking others, and the more nuanced choice models are assigning high probability to candidates when other candidates in their political party are removed.

## 6 Conclusion

We introduce the contextual repeated selection (CRS) model of ranking, a model that can eschew traditional assumptions such as intransitivty and unimodality allowing it to captures nuance in ranking. Our model fits data significantly better than existing models for a wide range of ranking domains including ranked choice voting, food preference surveys, race results, and search engine results. Our theoretical guarantees on the CRS model provide theoretical foundations for the performance we observe. Moreover, our risk analysis of ranking models closes gaps in the theory of maximum likelihood estimation for the multinomial logit (MNL) and Plackett-Luce (PL) models, and opens the door for future rich models and analyses of ranking data.

## Broader impact

Flexible ranking distributions that can be learned with provable guarantees can facilitate more powerful and reliable ranking algorithms inside recommender systems, search engines, and other ranking-based technological products. As a potential adverse consequence, more powerful and reliable learning algorithms can lead to an increased inappropriate reliance on technological solutions to complex problems, where practitioners may be not fully grasp the limitations of our work, e.g. independence assumptions, or that our risk bounds, as established here, do not hold for all data generating processes.

## Acknowledgements

This work is supported in part by an NSF Graduate Research Fellowship (AS), a Dantzig-Lieberman Fellowship and Krishnan Shah Fellowship (SR), a David Morgenthaler II Faculty Fellowship (JU), a Facebook Faculty Award (JU), a Young Investigator Award from the Army Research Office (73348-NS-YIP), and a gift from the Koret Foundation.

## Funding transparency statement

The funding sources supporting the work are described in the Acknowledgements section above. Over the past 36 months, AS has been employed part-time at StitchFix, held an internship at Facebook, and provided consulting services for JetBlue Technology Ventures. Over the past 36 months, SR has been employed at Twitter. Over the past 36 months, JU has received additional research funding from the National Science Foundation (NSF), the Army Research Office (ARO), a Hellman Faculty Fellowship, and the Stanford Thailand Research Consortium.

## Footnotes

[1]https://github.com/arjunsesh/lrr-neurips.

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
