[Supplementary Material]

## Supplemental material for "Learning Rich Rankings"

Arjun Seshadri, Stephen Ragain, Johan Ugander

The supplemental material is organized as follows. Appendix A gives further details of the datasets studied in the empirical analysis of the paper. Appendix B provides instructions for reproducing tables and plots in the paper. Appendix C provides additional simulation results. Appendix D gives proofs of the three main theorems of the paper. Appendix E gives proofs of auxiliary lemmas used in the proofs of the theorems.

## A Dataset descriptions

In our evaluation we study four widely studied datasets. All the datasets we study can be found in the Preflib repository[2]. First, the `sushi` dataset, consisting of 5,000 complete rankings of 10 types of sushi. Next, three election datasets, which consists of ranked choice votes given for three 2002 elections in Irish cities: the `dublin-north` election (abbreviated `dub-n` in tables) had 12 candidates and 43,942 votes for lists of varying length, `meath` had 14 candidates and 64,081 votes, and `dublin-west` (abbreviated `dub-w`) had 9 candidates and 29,988 votes. Third, the `nascar` dataset representing competitions, which consists of the partial ordering given by finishing drivers in each race of the 2002 Winston Cup. The data includes 74 drivers (alternatives) and 35 races (rankings).

The fourth collection we emphasize is the popular `LETOR` collection of datasets, which consists of ranking data arising from search engines. Although the `LETOR` data arises from algorithmic rather than human choices, it demonstrates the efficacy of our algorithms in large sparse data regimes. After removing datasets with fewer than 10 rankings and more than 100 alternatives (arbitrary thresholds that exclude small datasets with huge computational costs), the `LETOR` collection includes 727 datasets with a total of 12,838 rankings of between 3 and 50 alternatives.

Beyond these four emphasized collections, we include analyses of all 51 other Preflib datasets (as of May 2020) that contain partial or complete rankings of up to 10 items and at most 1000 rankings, a total of 11,956 rankings (these thresholds were again decided arbitrarily for computational reasons). We call this collection of datasets `PREF-SOI`, adopting the notation of [36]. We separately study the subset of 10 datasets comprised of complete rankings, referred to here-in as `PREF-SOC`, which contain a total of 5,116 rankings. The complete rankings in the `PREF-SOC` collection are suitable for both repeated selection and repeated elimination. While the sushi (complete ranking) and election (partial ranking) datasets are part of Preflib, they are comparatively quite large and are excluded from these two collections (`PREF-SOC` and `PREF-SOI`, respectively) by the above thresholds.

## B Reproducibility

Code that faithfully reproduces the Tables and Figures in both the main paper and the supplement is available at `https://github.com/arjunsesh/lrr-neurips`. See the Reproducibility Section of the `README` for details.

## C Simulation results

In this brief supplement we provide simulations that serve to validate our theoretical results. Figure 3 does so in two ways: first, showing that the error rate indeed decreases with $1/\ell$ as suggested by our risk bounds, and second that it does so with seemingly high probability, as shown by our tail bounds. The figure highlights three special cases, a PL model fit on PL data, a CRS model fit on PL data, and a CRS model fit on CRS data. All datasets consist of rankings of $n = 6$ items. For the PL model the number of parameters is $n = 6$. For the CDM model the number of parameters is $d = n(n-1) = 30$. In both cases, the model parameters were sampled from a truncated standard normal distribution within a $B$-ball with $B = 1.5$ (per the theorem statements). In all three panels, we generate 20 datasets from the underlying model, and fit cumulative increments 20 times to generate the result. The tight bundle that the 20 datasets form indicates how little the randomness of a given

Figure 3: Squared $\ell_2$ error of an estimated models in 20 growing datasets for the *(a)* PL estimation error on PL model data, *(b)* CRS estimation error on PL model data, and *(c)* CRS estimation error on CRS model data. The dashed black lines are a visual guide of the slope $1/\ell$, and the bundles represent the 20 different datasets; the tightness of the bundle validate our tail bounds.

dataset causes the risk to deviate. As in our main empirics, all maximum likelihood estimates were found using gradient-based optimization implemented in Pytorch.

In Figure 4 panel (a), we demonstrate simulations that suggest that the full CRS model's true convergence rate appears to be $O(n^2/\ell)$, as opposed to the larger $n$-dependence, $O(n^7/\ell)$, that we were able to guarantee theoretically in Theorem 3. We generate the plot in a manner similar to Figure 3, by generating 20 datasets and fitting them incrementally, this time averaging the performance of all 20 datasets to produce a single line per model. We repeat this process for four different model sizes $d = n(n-1)$ corresponding to $n \in \{6, 9, 12, 16\}$. We then plot the resulting risk multiplied by $\ell/n^2$. The apparently constant set of lines over the wide range of parameters and dataset sizes indicates that the risk of the full CRS model is likely close to $O(n^2/\ell)$ in theory, suggesting room for improvement in our analysis.

In Figure 4 panel (b), we demonstrate simulations that suggest that the true convergence rate of CRS models of various rank $r$ appears to be $O(nr/\ell)$, still smaller than the simulated rate for the full CRS model. The simulations are performed similarly to to those in panel (a), with the exception of plotting the risk multiplied by $\ell/nr$. As before, the apparent constant set of lines over the wide range of dataset sizes and parameters indicates that the risk of the CRS model is likely close to $O(nr/\ell)$ when $r \ll n$ (typical in practical settings), highlighting the practical value of the CRS for a small $r$.

Figure 4: Visualizations of the rate of CRS convergence. *(a)* Full CRS error multiplied by $\ell/n^2$. The legend highlights the parameters $d = n(n-1)$ of the different models. (b) CRS error multiplied by $\ell/nr$. The legend highlights the parameters $n$ and $r$ of the different models. The figure demonstrates that, over a wide range of parameters and rankings, the true rate of convergence for the full CRS is likely $O(n^2/\ell)$ while the rate for the CRS is likely $O(nr/\ell)$.

# D   Main proofs

## D.1   Proof of Theorem 1

**Theorem 1.** *Let $\theta^\star$ denote the true MNL model from which data is drawn. Let $\hat{\theta}_{MLE}$ denote the maximum likelihood solution. For any $\theta^\star \in \Theta_B = \{\theta \in \mathbb{R}^n : \|\theta\|_\infty \leq B, \mathbf{1}^T\theta = 0\}$, $t > 1$, and dataset $\mathcal{D}$ generated by the MNL model,*

$$\mathbb{P}\Big[ \left\| \hat{\theta}_{MLE}(\mathcal{D}) - \theta^\star \right\|_2^2 \geq c_{B,k_{max}} \frac{t}{m\lambda_2(L)^2} \Big] \leq e^{-t},$$

where $k_{max}$ is the maximum choice set size in $\mathcal{D}$, $c_{B,k_{max}}$ is a constant that depends on $B$ and $k_{max}$, and $\lambda_2(L)$ depends on the spectrum of a Laplacian $L$ formed by $\mathcal{D}$. For the expected risk,

$$\mathbb{E}\Big[\big\|\hat{\theta}_{MLE}(\mathcal{D}) - \theta^\star\big\|_2^2\Big] \leq c'_{B,k_{max}}\frac{1}{m\lambda_2(L)^2},$$

where $c'_{B,k_{max}}$ is again a constant that depends on $B$ and $k_{max}$.

**Proof.**

We are given some true MNL model with parameters $\theta^\star \in \Theta_B$, and for each datapoint $j \in [m]$ we have the probability of choosing item $x$ from set $C_j$ as

$$\mathbb{P}(y_j = x|\theta^\star, C_j) = \frac{\exp(\theta_x^\star)}{\sum_{y \in C_j}\exp(\theta_y^\star)}.$$

We will first introduction notation for analyzing the risk, and then proceed to first give a proof of the expected risk bound. We then carry the technology of that proof forward to give a proof of the tail bound statement.

**Notation.** We now introduce notation that will let us represent the above expression in a more compact manner. Because our datasets involve choice sets of multiple sizes, we use $k_j \in [k_{\min}, k_{\max}]$ to denote the choice set size for datapoint $j$, $|C_j|$. Extending a similar concept in [49] to the multiple set sizes, we then define matrices $E_{j,k_j} \in \mathbb{R}^{n \times k_j}$, $\forall j \in [m]$ as follows: $E_{j,k_j}$ has a column for every item $y \in C_j$ (and hence $k_j$ columns), and the column corresponding to item $y \in C_j$ simply has the $n$-dimensional unit vector $e_y$. This definition then renders the vector-matrix product $\theta^T E_{j,k_j} = [\theta_{y_1}, \theta_{y_2}, \theta_{y_3}, \ldots \theta_{y_{k_j}}] \in \mathbb{R}^{1 \times k_j}$.

Next, we define a collection of functions $F_k : \mathbb{R}^k \mapsto [0,1], \forall k \in [k_{\min}, k_{\max}]$ as

$$F_k([x_1, x_2, \ldots, x_k]) = \frac{\exp(x_1)}{\sum_{l=1}^k \exp(x_l)},$$

where the numerator always corresponds to the first entry of the input. These functions $F_k$ have several properties that will become useful later in the proof. First, it is easy to verify that all $F_k$ are shift-invariant, that is, $F_k(x) = F_k(x + c\mathbf{1})$, for any scalar $c$.

Next, from Lemma 1, we have that $\mathbf{1} \in \text{null}(\nabla^2(-\log(F_k(x))))$ and that

$$\nabla^2(-\log(F_k(x))) \succeq H_k = \beta_k(I - \frac{1}{k}\mathbf{1}\mathbf{1}^T), \tag{3}$$

where

$$\beta_k := \frac{1}{k\exp(2B)}. \tag{4}$$

That is, $F_k$ are strongly log-concave with a null space *only* in the direction of $\mathbf{1}$, since $\nabla^2(-\log(F_k(x))) \succeq H_k$ for some $H_k \in \mathbb{R}^{k \times k}$, $\lambda_2(H_k) > 0$.

As a final notational addition, in the same manner as [49] but accounting for multiple set sizes, we define $k$ permutation matrices $R_{1,k}, \ldots, R_{k,k} \in \mathbb{R}^{k,k}, \forall k \in [k_{\min}, k_{\max}]$, representing $k$ cyclic shifts in a fixed direction. Specifically, given some vector $x \in \mathbb{R}^k$, $y = x^T R_{l,k}$ is simply $x^T$ cycled (say, clockwise) so $y_1 = x_l$, $y_i = x_{(l+i-1)\%k}$, where $\%$ is the modulo operator. That is, these matrices allow for the cycling of the entries of row vector $v \in \mathbb{R}^{1 \times k}$ so that any entry can become the first entry of the vector, for any of the relevant $k$. This construction allows us to represent any choice made from the choice set $C_j$ as the first element of the vector $x$ that is input to $F$, thereby placing it in the numerator.

**First, an expected risk bound.** Given the notation introduced above, we can now state the probability of choosing the item $x$ from set $C_j$ compactly as:

$$\mathbb{P}(y_j = x|\theta^\star, C_j) = \mathbb{P}(y_j = x|\theta^\star, k_j, E_{j,k_j}) = F_{k_j}(\theta^{\star T}E_{j,k_j}R_{x,k_j}).$$

We can then rewrite the MNL likelihood as

$$\sup_{\theta \in \Theta_B} \prod_{(x_j, k_j, E_{j,k_j}) \in \mathcal{D}} F_{k_j}(\theta^T E_{j,k_j} R_{x_j, k_j}),$$

and the scaled negative log-likelihood as

$$\ell(\theta) = -\frac{1}{m} \sum_{(x_j, k_j, E_{j,k_j}) \in \mathcal{D}} \log(F_{k_j}(\theta^T E_{j,k_j} R_{x_j, k_j})) = -\frac{1}{m} \sum_{j=1}^{m} \sum_{i=1}^{k_j} \mathbf{1}[y_j = i] \log(F_{k_j}(\theta^T E_{j,k_j} R_{i,k_j})).$$

Thus,

$$\hat{\theta}_{\text{MLE}} = \arg\min_{\theta \in \Theta_B} \ell(\theta).$$

The compact notation makes the remainder of the proof a straightforward application of results from convex analysis: we first demonstrate that the scaled negative log-likelihood is strongly convex with respect to a semi-norm[3], and we use this property to show the proximity of the MLE to the optimal point as desired. The remainder of our expected risk bound proof mirrors that in [49] with a few extra steps of accounting created by the multiple set sizes. Beyond the additional accounting, one technical novelty in this expected risk proof, relative that in [49], is the development of Lemma 1 and its use to give a more careful handling of the Hessian. This handling is built on our observation that the Hessian is a rank-one modification of a symmetric matrix, whereby we can employ an argument due to Bunch–Nielsen–Sorensen [10] that relates the eigenvalues of such a matrix to the eigenvalues of its symmetric part. The tail bound proof (that follows this expected risk bound) is based on technical innovations that depart from previous strategies and will be surveyed there.

First, we have the gradient of the negative log-likelihood as

$$\nabla \ell(\theta) = -\frac{1}{m} \sum_{j=1}^{m} \sum_{i=1}^{k_j} \mathbf{1}[y_j = i] E_{j,k_j} R_{i,k_j} \nabla \log(F_{k_j}(\theta^T E_{j,k_j} R_{i,k_j})),$$

and the Hessian as

$$\nabla^2 \ell(\theta) = -\frac{1}{m} \sum_{j=1}^{m} \sum_{i=1}^{k_j} \mathbf{1}[y_j = i] E_{j,k_j} R_{i,k_j} \nabla^2 \log(F_{k_j}(\theta^T E_{j,k_j} R_{i,k_j})) R_{i,k_j}^T E_{j,k_j}^T.$$

We then have, for any vector $z \in \mathbb{R}^n$,

$$z^T \nabla^2 \ell(\theta) z = -\frac{1}{m} \sum_{j=1}^{m} \sum_{i=1}^{k_j} \mathbf{1}[y_j = i] z^T E_{j,k_j} R_{i,k_j} \nabla^2 \log(F_{k_j}(\theta^T E_{j,k_j} R_{i,k_j})) R_{i,k_j}^T E_{j,k_j}^T z$$

$$= \frac{1}{m} \sum_{j=1}^{m} \sum_{i=1}^{k_j} \mathbf{1}[y_j = i] z^T E_{j,k_j} R_{i,k_j} \nabla^2 (-\log(F_{k_j}(\theta^T E_{j,k_j} R_{i,k_j}))) R_{i,k_j}^T E_{j,k_j}^T z$$

$$\geq \frac{1}{m} \sum_{j=1}^{m} \sum_{i=1}^{k_j} \mathbf{1}[y_j = i] z^T E_{j,k_j} R_{i,k_j} H_{k_j} R_{i,k_j}^T E_{j,k_j}^T z$$

$$= \frac{1}{m} \sum_{j=1}^{m} \sum_{i=1}^{k_j} \mathbf{1}[y_j = i] z^T E_{j,k_j} R_{i,k_j} \frac{\beta_{k_j}}{k_j} (k_j I - \mathbf{1}\mathbf{1}^T) R_{i,k_j}^T E_{j,k_j}^T z$$

$$\geq \frac{1}{m} \sum_{j=1}^{m} \sum_{i=1}^{k_j} \mathbf{1}[y_j = i] z^T E_{j,k_j} \frac{\beta_{k_j}}{k_j} (k_j I - \mathbf{1}\mathbf{1}^T) E_{j,k_j}^T z$$

$$= \frac{1}{m} \sum_{j=1}^{m} \frac{\beta_{k_j}}{k_j} z^T E_{j,k_j} (k_j I - \mathbf{1}\mathbf{1}^T) E_{j,k_j}^T z$$

$$= \frac{\beta_{k_{\max}}}{m} \sum_{j=1}^{m} \frac{1}{k_j} z^T E_{j,k_j} (k_j I - \mathbf{1}\mathbf{1}^T) E_{j,k_j}^T z.$$

The first line follows from applying the definition of the Hessian. The second line follows from pulling the negative sign into the $\nabla^2$ term. The third and fourth line follow from Equation (3), strong log-concavity of all $F_k$. The fifth line follows recognizing that $H_k$ is invariant to permutation matrices. The sixth line follows from removing the inner sum since the terms are independent of $i$. The seventh line follows from lower bounding $\beta_{k_j}$ by $\beta_{k_{\max}}$.

Now, defining the matrix $L$ as

$$L = \frac{1}{m} \sum_{j=1}^{m} E_{j,k_j} (k_j I - \mathbf{1}\mathbf{1}^T) E_{j,k_j}^T,$$

we first note a few properties of $L$. First, it is easy to verify that $L$ is the Laplacian of a weighted graph on $n$ vertices, with each vertex corresponding to an item. This conclusion follows because each term in the average corresponds to the Laplacian of an unweighted clique on the subset of nodes $C_j$, and the average of unweighted Laplacians is a weighted graph Laplacian. Weighted edges of the graph represented by $L$ then denote when nonzero whether a pair of items has been compared in the dataset—that is, whether the pair of items has appeared together in some set $C_j$ for some datapoint $j$. The weights of the edges then denote the proportion of times the corresponding pairs have been compared in the dataset.

It is now easy to verify that $L\mathbf{1} = 0$, and hence $\text{span}(\mathbf{1}) \subseteq \text{null}(L)$. Moreover, we can show that $\lambda_2(L) > 0$, that is, $\text{null}(L) \subseteq \text{span}(\mathbf{1})$, as long as the weighted graph represented by $L$ is connected. This result follows because the number of zero eigenvalues of a weighted graph Laplacian represents the number of connected components of the graph. Hence, if the graph represented by $L$ is connected, then $\lambda_2(L) > 0$.

We also define the matrix

$$\hat{L} = \frac{1}{m} \sum_{j=1}^{m} \frac{1}{k_j} E_{j,k_j} (k_j I - \mathbf{1}\mathbf{1}^T) E_{j,k_j}^T.$$

Since $\frac{1}{k_j}$ is strictly positive, $\hat{L}$ has nonzero weighted edges exactly where the graph represented by $L$ does, but different weights. Hence, the two corresponding graphs' number of connected components are identical, and $\text{null}(\hat{L}) \subseteq \text{span}(\mathbf{1})$ if and only if $\text{null}(L) \subseteq \text{span}(\mathbf{1})$. Moreover, since $\hat{L} \succeq \frac{1}{k_{\max}} L$, we also have that $\lambda_2(\hat{L}) \geq \frac{1}{k_{\max}} \lambda_2(L)$. We work with $\hat{L}$ for the remainder of the proof, but state our final results in terms of the eigenvalues of $L$. We use $L$ in our results to maintain consistency of the final result with that of [49], and use $\hat{L}$ in our proof to produce sharper results for the multiple set size case.

With the matrix $\hat{L}$, we can write,

$$z^T \nabla^2 \ell(\theta) z \geq \beta_{k_{\max}} z^T \hat{L} z = \beta_{k_{\max}} ||z||_{\hat{L}}^2,$$

which is equivalent to stating that $\ell(\theta)$ is $\beta_{k_{\max}}$-strongly convex with respect to the $\hat{L}$ semi-norm at all $\theta \in \Theta_B$. Since $\theta^\star, \hat{\theta}_{\text{MLE}} \in \Theta_B$, strong convexity implies that

$$\beta_{k_{\max}} ||\hat{\theta}_{\text{MLE}} - \theta^\star||_{\hat{L}}^2 \leq \langle \nabla \ell(\hat{\theta}_{\text{MLE}}) - \nabla \ell(\theta^\star), \hat{\theta}_{\text{MLE}} - \theta^\star \rangle.$$

Further, we have

$$\begin{aligned}
\langle \nabla \ell(\hat{\theta}_{\text{MLE}}) - \nabla \ell(\theta^\star), \hat{\theta}_{\text{MLE}} - \theta^\star \rangle &= -\langle \nabla \ell(\theta^\star), \hat{\theta}_{\text{MLE}} - \theta^\star \rangle \\
&\leq |(\hat{\theta}_{\text{MLE}} - \theta^\star)^T \nabla \ell(\theta^\star)| \\
&= |(\hat{\theta}_{\text{MLE}} - \theta^\star)^T \hat{L}^{\frac{1}{2}} \hat{L}^{\frac{1}{2}\dagger} \nabla \ell(\theta^\star)| \\
&\leq ||\hat{L}^{\frac{1}{2}} (\hat{\theta}_{\text{MLE}} - \theta^\star)||_2 ||\hat{L}^{\frac{1}{2}\dagger} \nabla \ell(\theta^\star)||_2 \\
&= ||\hat{\theta}_{\text{MLE}} - \theta^\star||_{\hat{L}} ||\nabla \ell(\theta^\star)||_{\hat{L}^\dagger}.
\end{aligned}$$

Here the third line follows from the fact that $\mathbf{1}^T(\hat{\theta}_{\text{MLE}} - \theta^\star) = 0$, and so $(\hat{\theta}_{\text{MLE}} - \theta^\star) \perp \text{null}(\hat{L})$, which also implies that $(\hat{\theta}_{\text{MLE}} - \theta^\star) \perp \text{null}(\hat{L}^{\frac{1}{2}})$, and so $(\hat{\theta}_{\text{MLE}} - \theta^\star) \hat{L}^{\frac{1}{2}} \hat{L}^{\frac{1}{2}\dagger} = (\hat{\theta}_{\text{MLE}} - \theta^\star)$. The fourth line follows from Cauchy-Schwarz. Thus, we can conclude that

$$\beta_{k_{\max}}^2 ||\hat{\theta}_{\text{MLE}} - \theta^\star||_{\hat{L}}^2 \leq ||\nabla \ell(\theta^\star)||_{\hat{L}^\dagger}^2 = \nabla \ell(\theta^\star)^T \hat{L}^\dagger \nabla \ell(\theta^\star).$$

Now, all that remains is bounding the term on the right hand side. Recall the expression for the gradient

$$\nabla \ell(\theta^\star) = -\frac{1}{m} \sum_{j=1}^{m} \sum_{i=1}^{k_j} \mathbf{1}[y_j = i] E_{j,k_j} R_{i,k_j} \nabla \log(F_{k_j}(\theta^{\star T} E_{j,k_j} R_{i,k_j})) = -\frac{1}{m} \sum_{j=1}^{m} E_{j,k_j} V_{j,k_j},$$
(5)

where in the equality we have defined $V_{j,k_j} \in \mathbb{R}^{k_j}$ as

$$V_{j,k_j} := \sum_{i=1}^{k_j} \mathbf{1}[y_j = i] R_{i,k_j} \nabla \log(F_{k_j}(\theta^{\star T} E_{j,k_j} R_{i,k_j})).$$

Now, taking expectations over the dataset, we have,

$$
\begin{aligned}
\mathbb{E}[V_{j,k_j}] &= \mathbb{E}\Big[ \sum_{i=1}^{k_j} \mathbf{1}[y_j = i] R_{i,k_j} \nabla \log(F_{k_j}(\theta^{\star T} E_{j,k_j} R_{i,k_j})) \Big] \\
&= \sum_{i=1}^{k_j} \mathbb{E}\Big[ \mathbf{1}[y_j = i] \Big] R_{i,k_j} \nabla \log(F_{k_j}(\theta^{\star T} E_{j,k_j} R_{i,k_j})) \\
&= \sum_{i=1}^{k_j} F_{k_j}(\theta^{\star T} E_{j,k_j} R_{i,k_j}) R_{i,k_j} \nabla \log(F_{k_j}(\theta^{\star T} E_{j,k_j} R_{i,k_j})) \\
&= \sum_{i=1}^{k_j} R_{i,k_j} \nabla F_{k_j}(\theta^{\star T} E_{j,k_j} R_{i,k_j}) \\
&= \nabla_z \Big( \sum_{i=1}^{k_j} F_{k_j}(z^T R_{i,k_j}) \Big) = \nabla_z(1) = 0.
\end{aligned}
$$

Here, the third equality follows from applying the expectation to the indicator and retrieving the true probability. The fourth line follows from applying the definition of gradient of log, and the final line from performing a change of variables $z = \theta^{\star T} E_{j,k_j}$, pulling out the gradient and undoing the chain rule, and finally, recognizing that the expression sums to 1 for any $z$, thus resulting in a 0 gradient. We note that an immediate consequence of the above result is that $\mathbb{E}[V] = 0$, since $V$ is simply a concatenation of the individual $V_{j,k_j}$.

Next, we have

$$
\begin{aligned}
\mathbb{E}[\nabla \ell(\theta^\star)^T \hat{L}^\dagger \nabla \ell(\theta^\star)] &= \frac{1}{m^2} \mathbb{E}\Big[ \sum_{j=1}^{m} \sum_{l=1}^{m} V_{j,k_j}^T E_{j,k_j}^T \hat{L}^\dagger E_{l,k_l} V_{l,k_l} \Big] \\
&= \frac{1}{m^2} \mathbb{E}\Big[ \sum_{j=1}^{m} V_{j,k_j}^T E_{j,k_j}^T \hat{L}^\dagger E_{j,k_j} V_{j,k_j} \Big] \\
&\leq \frac{\lambda_n(\hat{L}^\dagger)}{m^2} \mathbb{E}\Big[ \sum_{j=1}^{m} V_{j,k_j}^T E_{j,k_j}^T E_{j,k_j} V_{j,k_j} \Big] \\
&= \frac{1}{m \lambda_2(\hat{L})} \mathbb{E}\Big[ \frac{1}{m} \sum_{j=1}^{m} V_{j,k_j}^T V_{j,k_j} \Big] \\
&\leq \frac{1}{m \lambda_2(\hat{L})} \mathbb{E}\Big[ \sup_{l \in [m]} ||V_{l,k_l}||_2^2 \Big],
\end{aligned}
$$

where the second line follows from the mean zero and independence of the $V_{j,k_j}$, the third from an upper bound of the quadratic form, the fourth from observing that the $E_{j,k_j}$ do not change the norm

of the $V_{j,k_j}$, and the last from averages being upper bound by maxima. We then have that,

$$\sup_{j\in[m]}||V_{j,k_j}||_2^2 = \sup_{j\in[m]}\sum_{i=1}^{k_j}\mathbf{1}[y_j=i]\nabla\log(F_{k_j}(\theta^T E_{j,k_j}R_{i,k_j}))^T R_{i,k_j}^T R_{i,k_j}\nabla\log(F_{k_j}(\theta^T E_{j,k_j}R_{i,k_j}))$$

$$= \sup_{j\in[m]}\sum_{i=1}^{k_j}\mathbf{1}[y_j=i]\nabla\log(F_{k_j}(\theta^T E_{j,k_j}R_{i,k_j}))^T\nabla\log(F_{k_j}(\theta^T E_{j,k_j}R_{i,k_j}))$$

$$= \sup_{j\in[m]}\sum_{i=1}^{k_j}\mathbf{1}[y_j=i]||\nabla\log(F_{k_j}(\theta^T E_{j,k_j}R_{i,k_j}))||_2^2$$

$$\leq \sup_{v\in[-(k_{\max}-1)B,(k_{\max}-1)B]^{k_{\max}}}||\nabla\log(F_{k_{\max}}(v))||_2^2 \leq 2,$$

where $R_{i,k_j}^T R_{i,k_j}$ in the first line is simply the identity matrix. For the final line, recalling the expression for the log gradient of $F_k$,

$$(\nabla\log(F_k(x)))_l = \mathbf{1}[l=1] - \frac{\exp(x_l)}{\sum_{p=1}^k \exp(x_p)},$$

it is straightforward to show that $\sup_{v\in[-(k_{\max}-1)B,(k_{\max}-1)B]^{k_{\max}}}||\nabla\log(F_{k_{\max}}(v))||_2^2$ is always upper bounded by 2.

Bringing this expression back to $\mathbb{E}[\nabla\ell(\theta^\star)^T\hat{L}^\dagger\nabla\ell(\theta^\star)]$, we have that

$$\mathbb{E}[\nabla\ell(\theta^\star)^T\hat{L}^\dagger\nabla\ell(\theta^\star)] \leq \frac{2}{m\lambda_2(\hat{L})}.$$

This expression in turn yields a bound on the expected risk in the $\hat{L}$ semi-norm, which is,

$$\mathbb{E}\left[\beta_{k_{\max}}^2||\hat{\theta}_{\text{MLE}}-\theta^\star||_{\hat{L}}^2\right] \leq \frac{2}{m\lambda_2(\hat{L})}.$$

By noting that $||\hat{\theta}_{\text{MLE}}-\theta^\star||_{\hat{L}}^2 = (\hat{\theta}_{\text{MLE}}-\theta^\star)\hat{L}(\hat{\theta}_{\text{MLE}}-\theta^\star) \geq \lambda_2(\hat{L})||\hat{\theta}_{\text{MLE}}-\theta^\star||_2^2$, since $\hat{\theta}_{\text{MLE}}-\theta^\star \perp$ null$(\hat{L})$, we can translate our finding into the $\ell_2$ norm:

$$\mathbb{E}\left[\beta_{k_{\max}}^2||\hat{\theta}_{\text{MLE}}-\theta^\star||_2^2\right] \leq \frac{2}{m\lambda_2(\hat{L})^2}.$$

Applying the fact that $\lambda_2(\hat{L}) \geq \frac{1}{k_{\max}}\lambda_2(L)$, we get:

$$\mathbb{E}[||\hat{\theta}_{\text{MLE}}-\theta^\star||_2^2] \leq \frac{2k_{\max}^2}{m\lambda_2(L)^2\beta_{k_{\max}}^2}.$$

Now, setting

$$c'_{B,k_{\max}} := \frac{2k_{\max}^2}{\beta_{k_{\max}}^2} = 2\exp(4B)k_{\max}^4,$$

we retrieve the expected risk bound in the theorem statement,

$$\mathbb{E}\left[\left\|\hat{\theta}_{\text{MLE}}(\mathcal{D})-\theta^\star\right\|_2^2\right] \leq c'_{B,k_{\max}}\frac{1}{m\lambda_2(L)^2}.$$

We close the expected risk portion of this proof with some remarks about $c_{B,k_{\max}}$. The quantity $\beta_{k_{\max}}$, defined in equation (4), serves as the important term that approaches 0 as a function of $B$ and $k_{\max}$, requiring that the former be bounded. Finally, $\lambda_2(L)$ is a parallel to the requirements on the algebraic connectivity of the comparison graph in [49] for the pairwise setting.

**From expected risk to tail bound.** Our proof of the tail bound is a continuation of the expected risk bound proof. While the expected risk bound closely followed the expected risk proof of [49], our tail bound proof contains significant novel machinery. Our presentation seem somewhat circular, given that the tail bound itself integrates out to an expected risk bound with the same parametric rates

(albeit worse constants), but we felt that to first state the expected risk bound was clearer, given that it arises as a stepping stone to the tail bound.

Recall again the expression for the gradient in Equation (5). Useful in our analysis will be an alternate expression:

$$\nabla \ell(\theta^\star) = -\frac{1}{m} \sum_{j=1}^{m} E_{j,k_j} V_{j,k_j} = -\frac{1}{m} E^T V,$$

where we have defined $V \in \mathbb{R}^{\Omega_{\mathcal{D}}}$ as the concatenation of all $V_{j,k_j}$, and $E \in \mathbb{R}^{\Omega_{\mathcal{D}} \times n}$, the vertical concatenation of all the $E_{j,k_j}$. Here, $\Omega_{\mathcal{D}} = \sum_{i=1}^{m} k_i$.

For the expected risk bound, we showed that $V_{j,k_j}$ have expectation zero, are independent, and $||V_{j,k_j}||_2^2 \leq 2$. Next, we have

$$(\nabla \log(F_k(x)))_l = \mathbf{1}[l = 1] - \frac{\exp(x_l)}{\sum_{p=1}^{k} \exp(x_p)}, \tag{6}$$

and so $\langle \nabla \log(F_k(x)), \mathbf{1} \rangle = \frac{1}{F_k(x)} \langle \nabla F_k(x), \mathbf{1} \rangle = \sum_{l=1}^{k} (\mathbf{1}[l = 1] - \frac{\exp(x_l)}{\sum_{p=1}^{k} \exp(x_p)}) = 0$, and hence, $V_{j,k_j}^T \mathbf{1} = 0$.

We now consider the matrix $M_k = (I - \frac{1}{k} \mathbf{1}\mathbf{1}^T)$. We note that $M_k$ has rank $k - 1$, with its nullspace corresponding to the span of the ones vector. We state the following identities:

$$M_k = M_k^\dagger = M_k^{\frac{1}{2}} = M_k^{\dagger \frac{1}{2}}.$$

Thus we have $M_{k_j} V_{j,k_j} = M_{k_j}^{\frac{1}{2}} M_{k_j}^{\frac{1}{2}} V_{j,k_j} = M_k M_k^\dagger V_{j,k_j} = V_{j,k_j}$, where the last equality follows since $V_{j,k_j}$ is orthogonal to the nullspace of $M_{k_j}$. We may now again revisit the expression for the gradient:

$$\nabla \ell(\theta^\star) = -\frac{1}{m} \sum_{j=1}^{m} E_{j,k_j} V_{j,k_j} = -\frac{1}{m} \sum_{j=1}^{m} E_{j,k_j} M_{k_j}^{1/2} V_{j,k_j} := -\frac{1}{m} X(\mathcal{D})^T V,$$

where we have defined $X(\mathcal{D}) \in \mathbb{R}^{\Omega_{\mathcal{D}} \times n}$ as the vertical concatenation of all the $E_{j,k_j} M_{k_j}^{1/2}$. As an aside, $X(\mathcal{D})$ is the design matrix in the terminology of generalized linear models (and is thus named fancifully).

Now, consider that

$$\nabla \ell(\theta^\star)^T \hat{L}^\dagger \nabla \ell(\theta^\star) = \frac{1}{m^2} V^T X(\mathcal{D}) \hat{L}^\dagger X(\mathcal{D})^T V.$$

We apply Lemma 3, a modified Hanson-Wright-type tail bound for random quadratic forms. This lemma follows from simpler technologies (largely Hoeffding's inequality) given that the random variables are bounded while also carefully handling the block structure of the problem.

In the language of Lemma 3 we have $V_{j,k_j}$ playing the role of $x^{(j)}$ and $\Sigma_{\mathcal{D}} := \frac{1}{m^2} X(\mathcal{D}) \hat{L}^\dagger X(\mathcal{D})^T$ plays the role of $A$. The invocation of this lemma is possible because $V_{j,k_j}$ is mean zero, $||V_{j,k_j}||_2 \leq \sqrt{2}$, and because $\Sigma_{\mathcal{D}}$ is positive semi-definite. We sweep $K^4 = 4$ from the lemma statement into the constant $c$ of the right hand side. Stating the result of Lemma 3 we have, for all $t > 0$,

$$\mathbb{P}(V^T \Sigma_{\mathcal{D}} V - \sum_{i=1}^{m} \lambda_{\max}(\Sigma_{\mathcal{D}}^{(i,i)}) \mathbb{E}[V^{(i)T} V^{(i)}] \geq t) \leq 2 \exp\left(-c \frac{t^2}{\sum_{i,j} \sigma_{\max}(\Sigma_{\mathcal{D}}^{(i,j)})^2}\right). \tag{7}$$

We note that

$$\sigma_{\max}(\Sigma_{\mathcal{D}}^{(i,j)}) = \sigma_{\max}\big(\frac{1}{m^2} M_{k_i}^{1/2} E_{i,k_i}^T \hat{L}^\dagger E_{j,k_j} M_{k_j}^{1/2}\big)$$

$$= \frac{1}{m^2} y_{\max}^T M_{k_i}^{1/2} E_{i,k_i}^T \hat{L}^\dagger E_{j,k_j} M_{k_j}^{1/2} z_{\max}$$

$$\leq \frac{1}{m^2} \lambda_{\max}(\hat{L}^\dagger) ||E_{i,k_i} M_{k_i}^{1/2} y_{\max}||_2 ||E_{j,k_j} M_{k_j}^{1/2} z_{\max}||_2$$

$$= \frac{1}{m^2} \lambda_{\max}(\hat{L}^\dagger) ||M_{k_i}^{1/2} y_{\max}||_2 ||M_{k_j}^{1/2} z_{\max}||_2$$

$$\leq \frac{1}{m^2 \lambda_2(\hat{L})},$$

for all $i, j$, where the second line follows because $y_{\max}$ and $z_{\max}$ are the maximum left and right singular vectors of unit norm, the third line from an upper bound on quadratic forms, the fourth because $E_{i,k_i}$ is a re-indexing that does not change Euclidean norm, and the final one because centering matrices can only lower the norm of a vector. This result has two consequences:

$$\lambda_{\max}(\Sigma_{\mathcal{D}}^{(i,i)}) = \sigma_{\max}(\Sigma_{\mathcal{D}}^{(i,i)}) \leq \frac{1}{m^2 \lambda_2(\hat{L})},$$

and

$$\sum_{i,j} \sigma_{\max}(\Sigma_{\mathcal{D}}^{(i,j)})^2 \leq \frac{m^2}{\lambda_2(\hat{L})^2 m^4} = \frac{1}{\lambda_2(\hat{L})^2 m^2}.$$

Now, noting that the norm of $V_{i,k_i}$ is bounded (thus $\mathbb{E}[V^{(i)T} V^{(i)}] \leq 2$), and substituting in the relevant values into Equation (7), we have for all $t > 0$:

$$\mathbb{P}\Big(\nabla\ell(\theta^\star)^T \hat{L}^\dagger \nabla\ell(\theta^\star) - \frac{2}{m\lambda_2(\hat{L})} \geq t\Big) \leq 2\exp\Big(-cm^2\lambda_2(\hat{L})^2 t^2\Big).$$

A variable substitution and simple algebra transforms this expression to

$$\mathbb{P}\left[\nabla\ell(\theta^\star)^T \hat{L}^\dagger \nabla\ell(\theta^\star) \geq c_2 \frac{t}{m\lambda_2(\hat{L})}\right] \leq e^{-t} \quad \text{for all } t > 1,$$

where $c_2$ is an absolute constant. We may then make the same substitutions as before with expected risk, to obtain,

$$\mathbb{P}\left[||\hat{\theta}_{\text{MLE}}(\mathcal{D}) - \theta^\star||_2^2 > c_2 \frac{t k_{\max}^2}{m\lambda_2(L)^2 \beta_{k_{\max}}^2}\right] \leq e^{-t} \quad \text{for all } t > 1.$$

Making the appropriate substitution with $c_{B,k_{\max}}$, we retrieve the second theorem statement, for another absolute constant $c$.

$$\mathbb{P}\left[\left\|\hat{\theta}_{\text{MLE}}(\mathcal{D}) - \theta^\star\right\|_2^2 \geq c_{B,k_{\max}} \frac{t}{m\lambda_2(L)^2}\right] \leq e^{-t} \quad \text{for all } t > 1.$$

Integrating the above tail bound leads to a similar bound on the expected risk (same parametric rates), albeit with a less sharp constants due to the added presence of $c$. $\qquad\square$

### D.2    Proof of Theorem 2

**Theorem 2.** *Let $\mathcal{R} = \sigma_1, \ldots, \sigma_\ell \sim PL(\theta^\star)$ be a dataset of full rankings generated from a Plackett-Luce model with true parameter $\theta^\star \in \Theta_B = \{\theta \in \mathbb{R}^n : \|\theta\|_\infty \leq B, \mathbf{1}^T \theta = 0\}$ and let $\hat{\theta}_{MLE}$ denote the maximum likelihood solution. Assume that $\ell > 4\log(\sqrt{\alpha_B}n)/\alpha_B^2$ where $\alpha_B$ is a constant that only depends on $B$. Then for $t > 1$ and any dataset $\mathcal{R}$ generated by the PL model,*

$$\mathbb{P}\left[\left\|\hat{\theta}_{MLE}(\mathcal{R}) - \theta^\star\right\|_2^2 \geq c_B'' \frac{n}{\ell} t\right] \leq e^{-t} + n^2 \exp(-\ell\alpha_B^2) \exp\Big(\frac{-t}{\alpha_B^2 n^2}\Big),$$

*where $c''_B$ is a constant that depends on $B$. For the expected risk,*

$$\mathbb{E}\Big[\Big\|\hat{\theta}_{MLE}(\mathcal{R}) - \theta^\star\Big\|_2^2\Big] \le c'_B \frac{n^3}{\ell}\mathbb{E}\Big[\frac{1}{\lambda_2(L)^2}\Big] \le c_B\frac{n}{\ell},$$

*where $c'_B = 4\exp(4B)$ and $c_B = 8\exp(4B)/\alpha_B^2$.*

**Proof.**

As with the proof for the MNL model, we first give an expected risk bound, and then proceed to carry that technology forward to give a tail bound. The tail bound will again integrate out to give an expected risk bound with the same parametric rates as the direct proof, albeit with weaker constants.

**Expected risk bound.** We exploit the fact that the PL likelihood is the MNL likelihood with $\ell(n-1)$ choices. We thus begin with the result of Theorem 1, unpacking $c_{B,k_{\max}}$ and applying $k_{\max} = n$ and $m = (n-1)\ell$:

$$\mathbb{E}[||\hat{\theta}_{\mathrm{MLE}} - \theta^\star||_2^2] \le \frac{2\exp(4B)n^4}{\ell(n-1)\lambda_2(L)^2}.$$

We remind the reader that since the choice sets are assumed fixed in the proof of 1, the expectation above is taken *only* over the choices, conditioned on the choice sets, and not over the choice sets themselves. Since we are now working with rankings, there is randomness over the choice sets themselves. The randomness manifests itself as an expectation *conditional* on the choice sets on the left hand side and in the randomness of $\lambda_2(L)$ on the right hand side. We may rewrite the expression to reflect this fact:

$$\mathbb{E}[||\hat{\theta}_{\mathrm{MLE}} - \theta^\star||_2^2 \mid S_1, S_2, ...S_{\ell(n-1)}] \le \frac{2\exp(4B)n^4}{\ell(n-1)\lambda_2(L)^2}.$$

and make progress towards the theorem statement, by take expectations over the choice sets $S_i$ on both sides and apply the law of iterated expectations:

$$\begin{aligned}
\mathbb{E}[||\hat{\theta}_{\mathrm{MLE}} - \theta^\star||_2^2] &= \mathbb{E}[\mathbb{E}[||\hat{\theta}_{\mathrm{MLE}} - \theta^\star||_2^2 \mid S_1, S_2, ...S_{\ell(n-1)}]] \\
&\le \mathbb{E}\Big[\frac{2\exp(4B)n^4}{\ell(n-1)\lambda_2(L)^2}\Big] \\
&= 4\exp(4B)\frac{n^3}{\ell}\mathbb{E}\Big[\frac{1}{\lambda_2(L)^2}\Big],
\end{aligned}$$

where in the last line we have bounded $n/(n-1)$ by 2. We have reached the intermediate form of the expected risk bound theorem statement.

What now remains is upper bounding $\mathbb{E}[1/\lambda_2(L)^2]$. Recall that $L$ is the Laplacian of a weighted comparison graph. A crude bound for $\lambda_2(L)$ comes from noting that choice set $\mathcal{X}$ appears at least $\ell$ times, each time adding $\frac{1}{\ell(n-1)}(nI - \mathbf{1}\mathbf{1}^T)$ to the Laplacian so that we get

$$\lambda_2(L) \ge \lambda_2(\frac{1}{n-1}(nI - \mathbf{1}\mathbf{1}^T)) = \lambda_2(\frac{n}{n-1}(I - \frac{1}{n}\mathbf{1}\mathbf{1}^T)) = \frac{n}{n-1} \ge 1 \qquad (8)$$

as $I - \frac{1}{n}\mathbf{1}\mathbf{1}^T$ is simply the centering matrix, and where the first inequality follows from properties of sums of PSD matrices [11][See pg. 128, Corollary (4.2)].

We will use a more sophisticated bound that comes from a careful study of the graph that the random Plackett-Luce Laplacian represents. We have packaged this analysis inside Lemma 4, which says that

$$\alpha_B n \le \lambda_2(L) \quad \text{with probability at least } 1 - n^2\exp\left(-\alpha_B^2\ell\right), \qquad (9)$$

where $\alpha_B = 1/(4(1 + 2e^{3B}))$.

We can use Lemma 4 to upper bound the expectation of $1/\lambda_2(L)^2$ as follows:

$$\mathbb{E}\Big[\frac{1}{\lambda_2(L)^2}\Big] = \mathbb{E}\Big[\frac{1}{\lambda_2(L)^2}\Big|\frac{1}{\lambda_2(L)} \le \frac{1}{\alpha_B n}\Big]\mathbb{P}\Big\{\frac{1}{\lambda_2(L)} \le \frac{1}{\alpha_B n}\Big\}$$

$$+ \mathbb{E}\Big[\frac{1}{\lambda_2(L)^2}\Big|\frac{1}{\lambda_2(L)} > \frac{1}{\alpha_B n}\Big]\mathbb{P}\Big\{\frac{1}{\lambda_2(L)} > \frac{1}{\alpha_B n}\Big\}$$

$$\le \frac{1}{\alpha_B^2 n^2} + \mathbb{P}\Big\{\frac{1}{\lambda_2(L)} > \frac{1}{\alpha_B n}\Big\}$$

$$\le \frac{1}{\alpha_B^2 n^2} + n^2 \exp\left(-\alpha_B^2 \ell\right),$$

where the first inequality follows from applying the bound of $1/(\alpha_B^2 n^2)$ to the first expectation and a bound of 1 to the second expectation (which comes from Equation (8)). The second inequality follows from applying the tail bound. Now, we need that $\ell > 4\log(\sqrt{\alpha_B}n)/\alpha_B^2$ to ensure that

$$\mathbb{E}\Big[\frac{1}{\lambda_2(L)^2}\Big] \le \frac{2}{\alpha_B^2 n^2}.$$

We can now circle back to the start of the proof to apply this result:

$$\mathbb{E}[||\hat{\theta}_{\text{MLE}} - \theta^\star||_2^2] \le 4\exp(4B)\frac{n^3}{\ell}\mathbb{E}\Big[\frac{1}{\lambda_2(L)^2}\Big] \le \frac{8\exp(4B)}{\alpha_B^2}\frac{n}{\ell},$$

so long as $\ell > 4\log(\sqrt{\alpha_B}n)/\alpha_B^2$. Defining $c_B$ as

$$c_B := \frac{8\exp(4B)}{\alpha_B^2},$$

we arrive at the expected risk bound in the theorem statement.

**Tail bound.** Our tail bound proof proceeds very similarly to that of the risk bound. To start, we again exploit the fact that the PL likelihood is the MNL likelihood with $\ell(n-1)$ choices. We thus begin with the result of Theorem 1, unpacking $c_{B,k_{\max}}$ and applying $k_{\max} = n$ and $m = (n-1)\ell$:

$$\mathbb{P}\left[||\hat{\theta}_{\text{MLE}}(\mathcal{D}) - \theta^\star||_2^2 > c_2 \frac{tn^3\exp(4B)}{\ell\lambda_2(L)^2}\right] \le e^{-t} \quad \text{for all } t > 1,$$

where $c_2$ is some absolute constant (note we have lower bounded $n/(n-1)$ by 2). Like before, we remind the reader that because the choice sets are assumed fixed in the proof of Theorem 1, the probabilistic statement *only* accounts for the randomness in the choices. Since we now are working with rankings, we must additionally account for the randomness over the choice sets, and the above statement is more clearly stated as a conditional probability over the sets:

$$\mathbb{P}\left[||\hat{\theta}_{\text{MLE}}(\mathcal{D}) - \theta^\star||_2^2 > c_2 \frac{tn^3\exp(4B)}{\ell\lambda_2(L)^2}\Big| S_1, S_2, ..., S_{\ell(n-1)}\right] \le e^{-t} \quad \text{for all } t > 1.$$

In order to obtain an unconditional statement, we now account for the choice sets $S$. Notice first that the expression depends only on the choice sets via the matrix $L$ (and more specifically its second smallest eigenvalue), and so:

$$\mathbb{P}\left[||\hat{\theta}_{\text{MLE}}(\mathcal{D})-\theta^\star||_2^2 > c_2 \frac{tn^3\exp(4B)}{\ell\lambda_2(L)^2}\Big|\lambda_2(L)\right] = \mathbb{P}\left[||\hat{\theta}_{\text{MLE}}(\mathcal{D})-\theta^\star||_2^2 > c_2 \frac{tn^3\exp(4B)}{\ell\lambda_2(L)^2}\Big|S_1, ..., S_{\ell(n-1)}\right]$$

We may additionally perform a change of variables, and rewrite the tail bound as

$$\mathbb{P}\left[||\hat{\theta}_{\text{MLE}}(\mathcal{D}) - \theta^\star||_2^2 > c_2 \frac{\delta n^3\exp(4B)}{\ell}\Big|\lambda_2(L)\right] \le e^{-\delta\lambda_2(L)^2} \quad \text{for all } \delta > 1/\lambda_2(L)^2. \quad (10)$$

We can now control $\lambda_2(L)$ using the same steps taken in the expected risk bound. Using Lemma 4, also stated above in (9), we can integrate Equation 10 over $\lambda_2(L)$:

$$\mathbb{E}_{\lambda_2(L)}\left[\mathbb{P}\left[||\hat{\theta}_{\text{MLE}}(\mathcal{D}) - \theta^\star||_2^2 > c_2 \frac{\delta n^3 \exp(4B)}{\ell}\Big|\lambda_2(L)\right]\right] \leq \mathbb{E}_{\lambda_2(L)}\left[\frac{1}{\exp(\delta\lambda_2(L)^2)}\right].$$

We may use the same trick to upper bound the right hand side just as we did the expectation of $1/\lambda_2(L)^2$ in the expected risk portion of our proof:

$$\mathbb{E}_{\lambda_2(L)}\left[\frac{1}{\exp(\delta\lambda_2(L)^2)}\right] = \mathbb{E}\left[\frac{1}{\exp(\delta\lambda_2(L)^2)}\Big|\frac{1}{\lambda_2(L)} \leq \frac{1}{\alpha_B n}\right]\mathbb{P}\left\{\frac{1}{\lambda_2(L)} \leq \frac{1}{\alpha_B n}\right\}$$

$$+ \mathbb{E}\left[\frac{1}{\exp(\delta\lambda_2(L)^2)}\Big|\frac{1}{\lambda_2(L)} > \frac{1}{\alpha_B n}\right]\mathbb{P}\left\{\frac{1}{\lambda_2(L)} > \frac{1}{\alpha_B n}\right\}$$

$$\leq \frac{1}{\exp(\delta\alpha_B^2 n^2)} + \frac{1}{\exp(\delta)}\mathbb{P}\left\{\frac{1}{\lambda_2(L)} > \frac{1}{\alpha_B n}\right\}$$

$$\leq \exp(-\delta\alpha_B^2 n^2) + \exp(-\delta)n^2\exp(-\alpha_B^2\ell),$$

where the first inequality follows from applying the bound of $1/\exp(\alpha_B^2 n^2)$ to the first expectation and a bound of $1/\exp(\delta)$ to the second expectation (which follows from Equation (8)). The second inequality follows from applying the tail bounds. Returning to the tail expression we have:

$$\mathbb{P}\left[||\hat{\theta}_{\text{MLE}}(\mathcal{D}) - \theta^\star||_2^2 > c_2 \frac{\delta n^3 \exp(4B)}{\ell}\right] \leq \exp(-\delta\alpha_B^2 n^2) + \exp(-\delta)\exp(-\ell\alpha_B^2).$$

Setting $t = \delta(\alpha_B^2 n^2)$, we obtain,

$$\mathbb{P}\left[||\hat{\theta}_{\text{MLE}}(\mathcal{D}) - \theta^\star||_2^2 > c_2 \frac{t n^3 \exp(4B)}{\ell\alpha_B^2 n^2}\right] \leq \exp(-t) + \exp\left(\frac{-t}{\alpha_B^2 n^2}\right)n^2\exp(-\ell\alpha_B^2) \quad \text{for all } t > 1.$$

Canceling terms we have,

$$\mathbb{P}\left[||\hat{\theta}_{\text{MLE}}(\mathcal{D}) - \theta^\star||_2^2 > c_2 \frac{\exp(4B)}{\alpha_B^2}\frac{n}{\ell}t\right] \leq e^{-t} + n^2\exp(-\ell\alpha_B^2)\exp\left(\frac{-t}{\alpha_B^2 n^2}\right) \quad \text{for all } t > 1.$$

Defining $c_B$ as

$$c_B := c_2 \frac{\exp(4B)}{\alpha_B^2},$$

we arrive at the tail bound in the theorem statement.

Integrating the above tail bound leads to a similar bound on the expected risk as the direct proof, albeit with less sharp constants.

### D.3    Proof of Theorem 3

**Theorem 3** *Let $\mathcal{R} = \sigma_1, \ldots, \sigma_\ell \sim CRS(u^\star)$ be a dataset of full rankings generated from the full CRS model with true parameter $u^\star \in \Theta_B = \{u \in \mathbb{R}^{n(n-1)} : u = [u_1^T, ..., u_n^T]^T; u_i \in \mathbb{R}^{n-1}, \|u_i\|_1 \leq B, \forall i; \mathbf{1}^T u = 0\}$ and let $\hat{u}_{MLE}$ denote the maximum likelihood solution. Assuming that $\ell > 8ne^{2B}\log(8ne^{2B})^2$, and $t > 1$:*

$$\mathbb{P}\left[||\hat{u}_{MLE}(\mathcal{R}) - u^\star||_2^2 > \frac{c_B''' n^7}{\ell}t\right] \leq e^{-t} + n\exp\left(-t\min\left\{\frac{c_B'' n^6}{\ell}, 1\right\}\right)e^{-\ell/(8ne^{2B})},$$

*where $c_B''$, $c_B'''$ are constants that depend only on $B$. For the expected risk,*

$$\mathbb{E}\left[||\hat{u}_{MLE}(\mathcal{R}) - u^\star||_2^2\right] \leq \mathbb{E}\left[\min\left\{\frac{c_B' n^3}{\ell\lambda_2(L)}, 4B^2 n\right\}\right] \leq c_B \frac{n^7}{\ell},$$

*where $c_B'$, $c_B$ are constants that depend only on $B$.*

**Proof of Theorem 3.**

As with the proof for the PL model, we first give an expected risk bound, and then proceed to carry that technology forward to give a tail bound. The tail bound will again integrate out to give an expected risk bound with the same parametric rates as the direct proof, albeit with weaker constants.

**Expected risk bound.** Our proof leverages the fact that the CRS likelihood is the CDM likelihood with $\ell(n-1)$ choices, just as our analysis of the PL model leveraged the relationship between the PL and MNL likelihoods. We thus begin with the result of Lemma 7, our adaptation of an existing CDM risk bound. Unpacking $c_{B,k_{\max}}$ and applying $k_{\max} = n$ and $m = (n-1)\ell$:

$$\mathbb{E}\left[||\hat{u}_{MLE}(\mathcal{D}) - u^*||_2^2\right] \leq \frac{2n(n-1)}{m\lambda_2(L)\beta_{k_{\max}}^2} \leq \frac{n^3(n-1)2\exp(4B)}{\ell(n-1)\lambda_2(L)} = \frac{n^3 2\exp(4B)}{\ell\lambda_2(L)}.$$

Working with rankings, we must handle the randomness over the choice sets themselves. The randomness manifests itself as an expectation *conditional* on the choice sets on the left hand side and in the randomness of $\lambda_2(L)$ on the right hand side. We may rewrite the expression in Lemma 7 to reflect this fact:

$$\mathbb{E}\left[||\hat{u}_{MLE}(\mathcal{D}) - u^*||_2^2 | S_1, S_2, ...S_{\ell(n-1)}\right] \leq \frac{n^3 2\exp(4B)}{\ell\lambda_2(L)}.$$

In Theorem 2 we proceeded to use a law of iterated expectations and then bound $\lambda_2(L)$. For the PL model, $\lambda_2(L)$ was always at least 1, and with high probability much larger. For the CRS model, however, $\lambda_2(L)$ can sometimes be 0. This result holds because non-trivial conditions on the choice set structure are required for the CDM model's *identifiability*. We refer the reader to [48] for more details. In our ranking setting, these conditions are never met with one ranking's worth of choices, and hence results in the CRS model parameters being underdetermined. As an aside, this claim should not be confused with the CRS model being unidentifiable. In fact, the CRS model with true parameter $u^\star \in \Theta_B$ is *always* identifiable That is, it is always determined with sufficiently many rankings, as we will later see.

Nevertheless, $1/\lambda_2(L)$ is difficult to meaningfully upper bound directly since $\lambda_2(L)$ can sometimes be 0. However, when the model is not identifiable the risk under our assumptions is *not* infinity. Because the true parameters live in a $\Theta_B$, a norm bounded space, we may bound the error of any guess $\hat{u}$ in that space:

$$||u^\star - \hat{u}||_2^2 = \sum_i ||u_i^\star - \hat{u}_i||_2^2 \leq \sum_i ||u_i^\star - \hat{u}_i||_1^2 \leq \sum_i (||u_i^\star||_1 + ||\hat{u}_i||_1)^2 \leq 4B^2 n.$$

We may thus bound the expected risk as

$$\mathbb{E}[||\hat{u}_{MLE} - u^*||_2^2 | S_1, S_2, ...S_{\ell(n-1)}] \leq \min\left\{\frac{2\exp(4B)n^3}{\ell\lambda_2(L)}, 4B^2 n\right\}.$$

and use a bound of $4B^2 n$ whenever $\lambda_2(L) = 0$. Now, we work towards the theorem statement by take expectations over the choice sets $S_i$ on both sides and apply the law of iterated expectations:

$$\mathbb{E}[||\hat{u}_{MLE} - u^*||_2^2] = \mathbb{E}[\mathbb{E}[||\hat{u}_{MLE} - u^*||_2^2 \mid S_1, S_2, ...S_{\ell(n-1)}]]$$
$$\leq \mathbb{E}\left[\min\left\{\frac{2\exp(4B)n^3}{\ell\lambda_2(L)}, 4B^2 n\right\}\right].$$

The above bound is the intermediate bound of the theorem statement, where $c_B' = 2\exp(4B)$.

We now use Lemma 5, which says that

$$\frac{1}{4n^3(n-1)e^{2B}} \leq \lambda_2(L) \quad \text{with probability at least } 1 - n\exp\left(-\frac{\ell}{8ne^{2B}}\right). \tag{11}$$

We can use Lemma 5 to upper bound the expectation of the risk as follows:

$$
\mathbb{E}\left[\min\left\{\frac{2\exp(4B)n^3}{\ell\lambda_2(L)}, 4B^2 n\right\}\right] = \mathbb{E}\left[\min\left\{\frac{2\exp(4B)n^3}{\ell\lambda_2(L)}, 4B^2 n\right\}\Big|\frac{1}{\lambda_2(L)}\le 4n^3(n-1)e^{2B}\right]
$$

$$
\times\, \mathbb{P}\left\{\frac{1}{\lambda_2(L)}\le 4n^3(n-1)e^{2B}\right\}
$$

$$
+\, \mathbb{E}\left[\min\left\{\frac{2\exp(4B)n^3}{\ell\lambda_2(L)}, 4B^2 n\right\}\Big|\frac{1}{\lambda_2(L)}> 4n^3(n-1)e^{2B}\right]
$$

$$
\times\, \mathbb{P}\left\{\frac{1}{\lambda_2(L)}> 4n^3(n-1)e^{2B}\right\}
$$

$$
\le \frac{2\exp(4B)n^3}{\ell}[4n^3(n-1)e^{2B}] + 4B^2 n\,\mathbb{P}\left\{\frac{1}{\lambda_2(L)}> 4n^3(n-1)e^{2B}\right\}
$$

$$
\le \frac{8\exp(6B)n^7}{\ell} + 4B^2 n^2 \exp\left(-\frac{\ell}{8ne^{2B}}\right),
$$

where the first inequality follows from selecting the first value in the min and applying the bound on $\lambda_2(L)$ to the first expectation; and a bound of $4B^2 n$ to the second expectation. The second inequality follows from applying the tail bound. Now, as long as $\ell > 8ne^{2B}\log(8ne^{2B})^2$, we may upper bound the second term as follows

$$
4B^2 n^2 \exp\left(-\frac{\ell}{8ne^{2B}}\right) \le \frac{4B^2 n^2}{\ell},
$$

and so

$$
\mathbb{E}[||\hat{u}_{MLE}-u^*||_2^2] \le \mathbb{E}\left[\min\left\{\frac{2\exp(4B)n^3}{\ell\lambda_2(L)}, 4B^2 n\right\}\right] \le \frac{8\exp(6B)n^7}{\ell}+\frac{4B^2 n^2}{\ell} \le \frac{12\exp(6B)n^7}{\ell},
$$

so long as $\ell > 8ne^{2B}\log(8ne^{2B})^2$, where the final inequality follows because $\exp(6B)/B^2 > 1$. Define $c_B := 12\exp(6B)$ and $c_B' := 2\exp(4B)$ to arrive at the theorem statement.

**Tail bound.** Our tail bound proof proceeds very similarly to that of the risk bound. To start, we again exploit the fact that the CRS likelihood is the CDM likelihood with $\ell(n-1)$ choices. We thus begin again with the result of Lemma 7, unpacking $c_{B,k_{\max}}$ and applying $k_{\max} = n$ and $m = (n-1)\ell$:

$$
\mathbb{P}\left[||\hat{u}_{\mathrm{MLE}}(\mathcal{D}) - u^\star||_2^2 > c_2\frac{tn^3\exp(4B)}{\ell\lambda_2(L)}\right] \le e^{-t} \quad \text{for all } t > 1.
$$

where $c_2$ is some absolute constant. Like before, we remind the reader that because the choice sets are assumed fixed in the proof of Lemma 7, the probabilistic statement *only* accounts for the randomness in the choices. Since we now are working with rankings, we must additionally account for the randomness over the choice sets, and the above statement is more clearly stated as a conditional probability over the sets:

$$
\mathbb{P}\left[||\hat{u}_{\mathrm{MLE}}(\mathcal{D}) - u^\star||_2^2 > c_2\frac{tn^3\exp(4B)}{\ell\lambda_2(L)}\Big| S_1, S_2, ..., S_{\ell(n-1)}\right] \le e^{-t} \quad \text{for all } t > 1.
$$

In order to obtain an unconditional statement, we now account for the choice sets $S$. Notice first that the expression depends only on the choice sets via the matrix $L$ (and more specifically its second smallest eigenvalue), and so:

$$
\mathbb{P}\left[||\hat{u}_{\mathrm{MLE}}(\mathcal{D})-u^\star||_2^2 > c_2\frac{tn^3\exp(4B)}{\ell\lambda_2(L)}\Big|\lambda_2(L)\right] = \mathbb{P}\left[||\hat{u}_{\mathrm{MLE}}(\mathcal{D})-u^\star||_2^2 > c_2\frac{tn^3\exp(4B)}{\ell\lambda_2(L)}\Big| S_1, ..., S_{\ell(n-1)}\right]
$$

Now, note additionally that

$$
\mathbb{P}\left[||\hat{u}_{\mathrm{MLE}}(\mathcal{D}) - u^\star||_2^2 > t4B^2 n\Big|\lambda_2(L)\right] = 0 \le e^{-t} \quad \text{for all } t \ge 1.
$$

and so

$$\mathbb{P}\left[||\hat{u}_{\text{MLE}}(\mathcal{D})-u^\star||_2^2 > t\frac{c_2 n^3 \exp(4B)}{\ell}\min\left\{\frac{1}{\lambda_2(L)},\frac{4B^2\ell}{c_2 n^2 \exp(4B)}\right\}\bigg|\lambda_2(L)\right] \le e^{-t} \ \text{ for all } t \ge 0.$$

We may additionally perform a change of variables, and rewrite the tail bound as

$$\mathbb{P}\left[||\hat{u}_{\text{MLE}}(\mathcal{D})-u^\star||_2^2 > c_2\frac{\delta n^3 \exp(4B)}{\ell}\bigg|\lambda_2(L)\right] \le e^{-\delta\max\left\{\lambda_2(L),\frac{c_2 n^2 \exp(4B)}{4B^2\ell}\right\}}. \qquad (12)$$

$$\text{for all } \delta > \min\left\{\frac{1}{\lambda_2(L)},\frac{4B^2\ell}{c_2 n^2 \exp(4B)}\right\}.$$

We can now control $\lambda_2(L)$ using the same steps taken in the expected risk bound. Using Lemma 5, also stated above in (11), we can integrate Equation (12) over $\lambda_2(L)$:

$$\mathbb{E}_{\lambda_2(L)}\left[\mathbb{P}\left[||\hat{u}_{\text{MLE}}(\mathcal{D})-u^\star||_2^2 > c_2\frac{\delta n^3 \exp(4B)}{\ell}\bigg|\lambda_2(L)\right]\right] \le \mathbb{E}_{\lambda_2(L)}\left[\frac{1}{\exp(\delta\max\left\{\lambda_2(L),\frac{c_2 n^2 \exp(4B)}{4B^2\ell}\right\})}\right].$$

We may use the same trick to upper bound the right hand side just as we did the expectation in the expected risk portion of our proof:

$$\mathbb{E}_{\lambda_2(L)}\left[\frac{1}{\exp(\delta\max\left\{\lambda_2(L),\frac{c_2 n^2 \exp(4B)}{4B^2\ell}\right\})}\right] = \mathbb{E}\left[\frac{1}{\exp(\delta\max\left\{\lambda_2(L),\frac{c_2 n^2 \exp(4B)}{4B^2\ell}\right\})}\bigg|\frac{1}{\lambda_2(L)} \le 4n^3(n-1)e^{2B}\right]$$

$$\times \mathbb{P}\left\{\frac{1}{\lambda_2(L)} \le 4n^3(n-1)e^{2B}\right\}$$

$$+ \mathbb{E}\left[\frac{1}{\exp(\delta\max\left\{\lambda_2(L),\frac{c_2 n^2 \exp(4B)}{4B^2\ell}\right\})}\bigg|\frac{1}{\lambda_2(L)} > 4n^3(n-1)e^{2B}\right]$$

$$\times \mathbb{P}\left\{\frac{1}{\lambda_2(L)} > 4n^3(n-1)e^{2B}\right\}$$

$$\le \frac{1}{\exp(\delta\max\left\{\frac{1}{4n^3(n-1)e^{2B}},\frac{c_2 n^2 \exp(4B)}{4B^2\ell}\right\})}$$

$$+ \frac{1}{\exp(\delta\frac{c_2 n^2 \exp(4B)}{4B^2\ell})}\mathbb{P}\left\{\frac{1}{\lambda_2(L)} > 4n^3(n-1)e^{2B}\right\}$$

$$\le \exp(-\delta\max\left\{\frac{1}{4n^3(n-1)e^{2B}},\frac{c_2 n^2 \exp(4B)}{4B^2\ell}\right\})$$

$$+ \exp(-\delta\frac{c_2 n^2 \exp(4B)}{4B^2\ell})n\exp\left(-\frac{\ell}{8ne^{2B}}\right),$$

where the first inequality follows from applying the bound on $1/\lambda_2(L)$ to the first expectation and a bound of the second term in the max to the second expectation. The second inequality follows from applying the tail bound probability from Lemma 5. Returning to the tail expression we have:

$$\mathbb{P}\left[||\hat{u}_{\text{MLE}}(\mathcal{D})-u^\star||_2^2 > c_2\frac{\delta n^3 \exp(4B)}{\ell}\right] \le \exp(-\delta\max\left\{\frac{1}{4n^3(n-1)e^{2B}},\frac{c_2 n^2 \exp(4B)}{4B^2\ell}\right\})$$

$$+ \exp(-\delta\frac{c_2 n^2 \exp(4B)}{4B^2\ell})n\exp\left(-\frac{\ell}{8ne^{2B}}\right).$$

Setting

$$t = \delta\max\left\{\frac{1}{4n^3(n-1)e^{2B}},\frac{c_2 n^2 \exp(4B)}{4B^2\ell}\right\},$$

we obtain

$$\mathbb{P}\left[||\hat{u}_{\mathrm{MLE}}(\mathcal{D}) - u^\star||_2^2 > t\min\left\{\frac{c_2\exp(6B)4n^6(n-1)}{\ell}, 4B^2n\right\}\right] \le$$

$$\exp(-t) + \exp\left(-t\min\left\{\frac{c_2\exp(6B)n^5(n-1)}{B^2\ell}, 1\right\}\right)n\exp\left(-\frac{\ell}{8ne^{2B}}\right) \quad \text{for all } t > 1.$$

Canceling terms we have, for all $t > 1$,

$$\mathbb{P}\left[||\hat{u}_{\mathrm{MLE}}(\mathcal{D}) - u^\star||_2^2 > t\frac{c_2\exp(6B)4n^7}{\ell}\right] \le e^{-t} + n\exp\left(-t\min\left\{\frac{c_2\exp(6B)n^6}{B^2\ell}, 1\right\}\right)e^{-\ell/(8ne^{2B})}.$$

Defining $c_B''$, $c_B'''$ as

$$c_B'' := c_2\frac{\exp(6B)}{B^2}, c_B''' := 4c_2\exp(6B),$$

we arrive at the tail bound in the theorem statement:

$$\mathbb{P}\left[||\hat{u}_{\mathrm{MLE}}(\mathcal{D}) - u^\star||_2^2 > \frac{c_B'''n^7}{\ell}t\right] \le e^{-t} + n\exp\left(-t\min\left\{\frac{c_B''n^6}{\ell}, 1\right\}\right)e^{-\ell/(8ne^{2B})}.$$

Integrating the above tail bound leads to a similar bound on the expected risk as the direct proof, albeit with less sharp constants. □

# E Proofs of auxiliary lemmas

**Lemma 1.** *For the collection of functions $F_k : \mathbb{R}^k \mapsto [0, 1]$, $\forall k \geq 2$ defined as*

$$F_k([x_1, x_2, \ldots, x_k]) = \frac{\exp(x_1)}{\sum_{l=1}^{k} \exp(x_l)},$$

*where $x \in [-B, B]^k$, we have that*

$$\nabla^2(-\log(F_k(x)))\mathbf{1} = 0,$$

*and*

$$\nabla^2(-\log(F_k(x))) \succeq \frac{1}{k \exp(2B)}(I - \frac{1}{k}\mathbf{1}\mathbf{1}^T).$$

**Proof.** We first compute the Hessian as:

$$\nabla^2(-\log(F_k(x))) = \frac{1}{(\langle \exp(x), 1 \rangle)^2}(\langle \exp(x), 1 \rangle \text{diag}(\exp(x)) - \exp(x)\exp(x)^T),$$

where $\exp(x) = [e^{x_1}, \ldots, e^{x_k}]$. Note that

$$
\begin{aligned}
v^T \nabla^2(-\log(F_k(x)))v &= \frac{1}{(\langle \exp(x), 1 \rangle)^2} v^T (\langle \exp(x), 1 \rangle \text{diag}(\exp(x)) - \exp(x)\exp(x)^T)v \\
&= \frac{1}{(\langle \exp(x), 1 \rangle)^2}(\langle \exp(x), 1 \rangle \langle \exp(x), v^2 \rangle - \langle \exp(x), v \rangle^2) \\
&\geq 0,
\end{aligned}
$$

where $v^2$ refers to the element-wise square operation on vector $v$. While the final inequality is an expected consequence of the positive semidefiniteness of the Hessian, we note that it also follows from an application of Cauchy-Schwarz to the vectors $\sqrt{\exp(x)}$ and $\sqrt{\exp(x)} \odot v$, and is thus an equality *if and only if* $v \in \text{span}(\mathbf{1})$. Thus, we have that the smallest eigenvalue $\lambda_1(\nabla^2(-\log(F_k(x)))) = 0$ is associated with the vector $\mathbf{1}$, a property we expect from shift invariance, and that the second smallest eigenvalue $\lambda_2(\nabla^2(-\log(F_k(x)))) > 0$. Thus, we can state that

$$\nabla^2(-\log(F_k(x))) \succeq H_k = \beta_k(I - \frac{1}{k}\mathbf{1}\mathbf{1}^T), \tag{13}$$

where

$$\beta_k := \min_{x \in [-B,B]^k} \lambda_2(\nabla^2(-\log(F_k(x)))), \tag{14}$$

and it's clear that $\beta_k > 0$. The minimization is taken over $x \in [-B, B]^k$ since each $x_i$ is simply an entry of the $\theta$ vector, each entry of which is in $[-B, B]$. We next show that

$$\beta_k \geq \frac{1}{k \exp(2B)},$$

to complete the result.

First, a definition:

$$p(x) := \frac{\exp(x)}{\langle \exp(x), 1 \rangle}.$$

Evidently, $p(x) \in \Delta_k$, and since $x \in [-B, B]^k$, $p(x) \succ 0$. We may also write the Hessian as

$$\nabla^2(-\log(F_k(x))) = \text{diag}(p(x)) - p(x)p(x)^T.$$

In this format, we may now directly apply Theorem 1 from [10], which lower bounds the second eigenvalue of the Hessian by the minimum probability in $p(x)$. Thus,

$$\lambda_2(\nabla^2(-\log(F_k(x)))) \geq \min_i p(x)_i$$

A simple calculation reveals then that

$$
\begin{aligned}
\beta_k = \min_{x \in [-B,B]^k} \lambda_2(\nabla^2(-\log(F_k(x)))) &\geq \min_{x \in [-B,B]^k} \min_i p(x)_i \\
&= \frac{1}{1 + (k-1)\exp(2B)} \\
&\geq \frac{1}{k\exp(2B)},
\end{aligned}
$$

which completes the proof.

$\square$

**Lemma 2.** *For $\Sigma_\mathcal{D} := \frac{1}{m^2} X(\mathcal{D}) \hat{L}^\dagger X(\mathcal{D})^T$, where the constituent quantities are defined in the proof of Lemma 7, we have,*

$$
\boldsymbol{tr}(\Sigma_\mathcal{D}) = \frac{d-1}{m}, \qquad\qquad ||\Sigma_\mathcal{D}||_F^2 = \frac{d-1}{m^2}.
$$

**Proof.** Consider first that

$$
\frac{1}{m} X(\mathcal{D})^T X(\mathcal{D}) = \frac{1}{m} \sum_{j=1}^m E_{j,k_j} M_{k_j}^{1/2} M_{k_j}^{1/2} E_{j,k_j}^T = \frac{1}{m} \sum_{j=1}^m E_{j,k_j}(I - \frac{1}{k_j}\mathbf{1}\mathbf{1}^T)E_{j,k_j}^T = \hat{L}.
$$

Since $\hat{L}$ is symmetric and positive semidefinite, it has an eigenvalue decomposition of $U\Lambda U^T$. By definition, the Moore-Penrose inverse is $\hat{L}^\dagger = U\Lambda^\dagger U^T$. We must have that $X(\mathcal{D}) = \sqrt{m}V\Lambda^{\frac{1}{2}}U^T$ for some orthogonal matrix $V$ in order for $\hat{L}$ to equal $\frac{1}{m} X(\mathcal{D})^T X(\mathcal{D})$. With these facts, we have

$$
\begin{aligned}
\frac{1}{m^2} X(\mathcal{D}) \hat{L}^\dagger X(\mathcal{D})^T &= \frac{1}{m^2} \sqrt{m} V\Lambda^{\frac{1}{2}} U^T U\Lambda^\dagger U^T U\Lambda^{\frac{1}{2}} V^T \sqrt{m} \\
&= \frac{1}{m} V\Lambda\Lambda^\dagger V^T.
\end{aligned}
$$

That is, $\Sigma_\mathcal{D}$ is a positive semi-definite matrix with spectra corresponding to $d-1$ values equaling $\frac{1}{m}$, and the last equaling 0. The result about the trace immediately follows. The equality about the Frobenius norm comes from the observation that the Frobenius norm of a positive semi-definite matrix is the squared sum of its eigenvalues. $\square$

**Lemma 3.** *Suppose we have a collection of mean zero independent random vectors $X^{(i)} \in \mathbb{R}^{k_i}$, $i = 1, \ldots, m$, of bounded Euclidean norm $||X^{(i)}||_2 \leq K$ stacked together into a single vector $X \in \mathbb{R}^d$, where $d = \sum_{i=1}^m k_i$. Additionally suppose we have a real positive semidefinite matrix $A \in \mathbb{R}^{d \times d}$ and denote by $A^{(i,j)} \in \mathbb{R}^{k_i \times k_j}$ the submatrix of $A$ whose rows align with the index position of $X^{(i)}$ in $X$ and whose columns align with the index position of $X^{(j)}$ in $X$. Then, for every $t \geq 0$,*

$$
\mathbb{P}(X^T A X - \sum_{i=1}^m \lambda_{max}(A^{(i,i)})\mathbb{E}[X^{(i)T} X^{(i)}] \geq t) \leq 2\exp\Big(-c\frac{t^2}{K^4 \sum_{i,j} \sigma_{max}(A^{(i,j)})^2}\Big),
$$

*where $\lambda_{max}(\cdot), \sigma_{max}(\cdot)$ refer to the maximum eigenvalue/singular value of a matrix, $||\cdot||_F$ and $||\cdot||_{op}$ respectively refer to the Frobenius and operator norm of a matrix, and $c > 0$ is an absolute positive constant.*

**Proof.** We heavily reference preliminary concepts about sub-Gaussian random variables as they are described in [52]. We assume without loss of generality that $K = 1$ ($||X^{(i)}||_2 \leq 1$), since we may substitute $X/K$ in place of any $X$ to satisfy the assumption, and rearrange terms to produce the result for any $K$.

First, note that

$$X^T A X - \sum_{i=1}^{m} \lambda_{\max}(A^{(i,i)})\mathbb{E}[X^{(i)T}X^{(i)}] = \sum_{i,j} X^{(i)T}A^{(i,j)}X^{(j)} - \sum_{i=1}^{m} \lambda_{\max}(A^{(i,i)})\mathbb{E}[X^{(i)T}X^{(i)}]$$

$$= \sum_{i} X^{(i)T}A^{(i,i)}X^{(i)} - \sum_{i=1}^{m} \lambda_{\max}(A^{(i,i)})\mathbb{E}[X^{(i)T}X^{(i)}]$$

$$+ \sum_{i,j\neq i} X^{(i)T}A^{(i,j)}X^{(j)}$$

$$\leq \sum_{i=1}^{m} \lambda_{\max}(A^{(i,i)})(X^{(i)T}X^{(i)} - \mathbb{E}[X^{(i)T}X^{(i)}])$$

$$+ \sum_{i,j,\neq i} X^{(i)T}A^{(i,j)}X^{(j)}.$$

Thus,

$$\mathbb{P}(X^T A X - \sum_{i=1}^{m} \lambda_{\max}(A^{(i,i)})\mathbb{E}[X^{(i)T}X^{(i)}] \geq t) \leq \mathbb{P}\Big( \sum_{i=1}^{m} \lambda_{\max}(A^{(i,i)})(X^{(i)T}X^{(i)} - \mathbb{E}[X^{(i)T}X^{(i)}])$$

$$+ \sum_{i,j\neq i} X^{(i)T}A^{(i,j)}X^{(j)} \geq t \Big),$$

and we may upper bound the right hand side to obtain an upper bound on the left hand side. We will in fact individually bound from above an expression corresponding to the block diagonal entries

$$p_1 = \mathbb{P}\Big( \sum_{i=1}^{m} \lambda_{\max}(A^{(i,i)})(X^{(i)T}X^{(i)} - \mathbb{E}[X^{(i)T}X^{(i)}]) \geq \frac{t}{2} \Big),$$

and an expression corresponding to the off block diagonal entries

$$p_2 = \mathbb{P}\Big( \sum_{i\neq j} X^{(i)T}A^{(i,j)}X^{(j)} \geq \frac{t}{2} \Big),$$

and use the union bound to obtain the desired result.

Before we proceed, we remind the reader that a random $Z$ is sub-Gaussian if and only if

$$(\mathbb{E}|Z|^p)^{1/p} \leq K_2\sqrt{p}, \ \forall p \geq 1,$$

where $K_2$ is a positive constant. We define the sub-Gaussian norm of a random variable, $||Z||_{\psi_2}$, as the smallest $K_2$ that satisfies the above expression, i.e.,

$$||Z||_{\psi_2} = \sup_{p\geq 1} p^{-1/2}(\mathbb{E}|Z|^p)^{1/p}.$$

The sub-Gaussian norm recovers the familiar bound on the moment generating function for centered $Z$:

$$\mathbb{E}[\exp(tZ)] \leq \exp(Ct^2||Z||_{\psi_2}), \ \forall t,$$

where $C$ is an absolute positive constant. Similarly, we have that a random variable $W$ is sub-exponential if and only if

$$(\mathbb{E}|W|^p)^{1/p} \leq K_2 p, \ \forall p \geq 1,$$

and similarly sub-exponential norm $||W||_{\psi_1}$ is defined as

$$||W||_{\psi_1} = \sup_{p\geq 1} p^{-1}(\mathbb{E}|Z|^p)^{1/p}.$$

**Part 1: The Block Diagonal Entries.** Define $y_i := ||X^{(i)}||_2^2$ and $a_i = \lambda_{\max}(A^{(i,i)})$, and let the vector $a \in \mathbb{R}^m$ consist of elements $a_i$. Recall that the norms of all $X^{(i)}$ are bounded by $K$ by assumption and $K = 1$, WLOG. Thus all $y_i$ are sub-Gaussian where $||y_i||_{\psi_2} \leq K^2 \leq 1$. Using this

notation we may then write the expression corresponding to the block diagonal entries in a more compact manner:

$$\sum_{i=1}^{m} \lambda_{\max}(A^{(i,i)})(X^{(i)T}X^{(i)} - \mathbb{E}[X^{(i)T}X^{(i)}]) = \sum_{i=1}^{m} a_i(y_i - \mathbb{E}[y_i]).$$

Note that centering does not change subgaussianity, and that $y_i^2 - \mathbb{E}[y_i^2]$ is a centered sub-gaussian random variable where

$$||y_i - \mathbb{E}[y_i]||_{\psi_2} \leq 2||y_i||_{\psi_2} \leq 2.$$

These inequalities follow from Remark 5.18 in [52]. Hence, we may apply a one-sided Hoeffding-type inequality (Proposition 5.10 in [52]) to state that

$$\mathbb{P}\Big( \sum_{i=1}^{m} a_i(y_i - \mathbb{E}[y_i]) \geq t \Big) \leq \exp\Big[ -c_1\Big(\frac{t^2}{4||a||_2^2}\Big)\Big],$$

where $c_1$ is some absolute positive constant.

Now, examining $||a||_2^2$ we see that

$$||a||_2^2 = \sum_{i=1}^{m} \lambda_{\max}(A^{(i,i)})^2 = \sum_{i=1}^{m} \sigma_{\max}(A^{(i,i)})^2 \leq \sum_{i,j} \sigma_{\max}(A^{(i,j)})^2 := \sigma. \qquad (15)$$

Thus assembling all the pieces together and lumping together absolute positive constants, we can conclude,

$$p_1 = \mathbb{P}\Big( \sum_{i=1}^{m} \lambda_{\max}(A^{(i,i)})(X^{(i)T}X^{(i)} - \mathbb{E}[X^{(i)T}X^{(i)}]) \geq \frac{t}{2} \Big) \leq \exp\Big[ -c_2\Big(\frac{t^2}{\sigma}\Big)\Big],$$

where $c_2$ is another absolute positive constant.

**Part 2: The Off Block Diagonal Entries.** In this section, we are attempting to bound:

Note that

$$|X^{(i)T}A^{(i,j)}X^{(j)}| \leq \sigma_{\max}(A^{(i,j)})||X^{(i)}||_2||X^{(j)}||_2.$$

Thus, since $||X^{(i)}|| \leq K = 1$, $X^{(i)T}A^{(i,j)}X^{(j)}$ is a mean zero sub-Gaussian random variable for all $i \neq j$ with sub-Gaussian norm:

$$||X^{(i)T}A^{(i,j)}X^{(j)}||_{\psi_2} \leq \sigma_{\max}(A^{(i,j)})K^2 \leq \sigma_{\max}(A^{(i,j)})$$

We may then again apply a one sided Hoeffding-type inequality to state that

$$\mathbb{P}\Big( \sum_{i \neq j} X^{(i)T}A^{(i,j)}X^{(j)} \geq t \Big) \leq \exp\Big[ -c_1\Big(\frac{t^2}{\sum_{i \neq j} \sigma_{\max}(A^{(i,j)})^2}\Big)\Big].$$

Now, we have

$$\sum_{i \neq j} \sigma_{\max}(A^{(i,j)})^2 \leq \sum_{i,j} \sigma_{\max}(A^{(i,j)})^2 = \sigma, \qquad (16)$$

Thus assembling all the pieces together and lumping together absolute positive constants, we can conclude,

$$p_2 = \mathbb{P}\Big( \sum_{i \neq j} X^{(i)T}A^{(i,j)}X^{(j)} \geq \frac{t}{2} \Big) \leq \exp\Big[ -c_3\Big(\frac{t^2}{\sigma}\Big)\Big],$$

This bound on $p_2$ is the same, up to the constant, as the bound on $p_1$ from Part 1.

To finish the proof, we can just use the union bound over the block diagonal and block off-diagonal results and lump together constants:

$$\mathbb{P}\Big( \sum_{i=1}^{m} \lambda_{\max}(A^{(i,i)})(X^{(i)T}X^{(i)} - \mathbb{E}[X^{(i)T}X^{(i)}]) + \sum_{i,j \neq i} X^{(i)T}A^{(i,j)}X^{(j)} \geq t \Big) \leq p_1 + p_2$$

$$\leq 2\exp\Big[ -c\Big(\frac{t^2}{\sigma_{\max}(A^{(i,j)})^2}\Big)\Big],$$

where $c$ is an absolute positive constant. Since the left hand side upper bounds the left hand side of the expression in the lemma statement, we can conclude the proof. $\qquad \square$

**Lemma 4.** *Let $\sigma^{(1)}, \sigma^{(2)}, ...\sigma^{(\ell)}$ be drawn iid $PL(\theta^\star)$, for some parameter $\theta^\star \in \Theta_B = \{\theta \in \mathbb{R}^d : \|\theta\|_\infty \leq B, \mathbf{1}^T \theta = 0\}$. Let $\lambda_2(L)$ be the second smallest eigenvalue of the (random) Plackett-Luce Laplacian obtained from $\ell$ samples of $PL(\theta^*)$. Then*

$$\alpha_B n \leq \lambda_2(L),$$

*with probability at least $1 - n^2 \exp\left(-\alpha_B^2 \ell\right)$, where $\alpha_B = 1/(4(1 + 2e^{3B}))$.*

**Proof.** The edge weights of the comparison graph, denoted by $\bar{w}_{ij}, \forall i \neq j$, for a collection of $\ell$ rankings can be described as:

$$\bar{w}_{ij} = \frac{1}{(n-1)} \frac{1}{\ell} \sum_{k=1}^{\ell} \min\{\sigma_k(i), \sigma_k(j)\}.$$

Where $L_{ij} = -\bar{w}_{ij}, \forall i \neq j$. The implication of considering the edge weights of the graph represented by $L$ is that we can lower bound $\lambda_2(L)$ in a simple way:

$$\lambda_2(L) \geq \lambda_2(K_n) \min_{ij} \bar{w}_{ij} = n \min_{ij} \bar{w}_{ij} = \frac{n}{n-1} \min_{ij} \left[\frac{1}{\ell} \sum_{k=1}^{\ell} \min\{\sigma_k(i), \sigma(j)\}\right]. \quad (17)$$

For the inequality, we use the fact that the algebraic connectivity of a graph on $n$ vertices $G$ must be lower bounded the algebraic connectivity of a complete graph on $n$ vertices $K_n$ whose edge weights are the smallest edge weight of $G$. The equality follows from the algebraic connectivity of a complete graph $K_n$.

We now need to unpack the right hand side of this bound. For each alternative pair $i, j \in \mathcal{X}$ and a ranking $\sigma \sim PL(\theta^\star)$, define the random variables $X_{ij} = \min\{\sigma(i), \sigma(j)\}$. Extend this notation so that for $k = 1, \ldots, \ell$, let $X_{ij}^{(k)} = \min\{\sigma(i)^{(k)}, \sigma(j)^{(k)}\}$ and additionally,

$$\bar{X}_{ij} = \frac{1}{\ell} \sum_{k=1}^{\ell} X_{ij}^{(k)} = \frac{1}{\ell} \sum_{k=1}^{\ell} \min\{\sigma(i)^{(\ell)}, \sigma(j)^{(\ell)}\}.$$

We are thus aiming to show that

$$\frac{1}{4}\left(\frac{n}{1 + 2e^{3B}} + 2\right) \leq \min_{i,j \in \mathcal{X}} \bar{X}_{ij}.$$

Intuitively, $\bar{X}_{ij}$ is the mean of $\ell$ iid variables $X_{ij}$, for each pair $i, j$. We have that $P(X_{ij} \geq k)$ is the probability that neither $i$ nor $j$ were chosen in the first $k - 1$ MNL choices of repeated selection. Of course, $P(X_{ij} \geq 1) = 1$ and $P(X_{ij} \geq n) = 0$. For $k = 2, \ldots, n-1$, we have,

$$P(X_{ij} \geq k) = \underbrace{\left(\sum_{(i_1,...,i_{k-2}) \in \mathcal{X} \backslash \{i,j\}} \prod_{m=1}^{k-2} \frac{e^{\theta_{i_m}}}{\sum_{x \in \mathcal{X} \backslash \cup_{q=1}^{m-1} \{i_q\}} e^{\theta_x}}\right)}_{\text{Neither } i \text{ nor } j \text{ are choice } 1, ..., k-2} \underbrace{\left(1 - \frac{e^{\theta_i} + e^{\theta_j}}{\sum_{x \in \mathcal{X} \backslash \{i_1,...i_{k-2}\}} e^{\theta_x}}\right)}_{\text{Neither } i \text{ nor } j \text{ are choice } k-1},$$

where we define the terms so that the first underbrace term just becomes 1 when $k = 2$.

We will now lower bounded the probability of not choosing item $i$ or $j$ in steps $1, ..., k-1$ as the probability of not choosing two items each with utility $B$ when all other items have utility $-B$. We

proceed as follows:

$$P(X_{ij} \geq k) = \sum_{(i_1,\ldots,i_{k-2}) \in \mathcal{X}\setminus\{i,j\}} \prod_{m=1}^{k-2} \frac{e^{\theta_{i_m}}}{\sum_{x \in \mathcal{X}\setminus \cup_{q=1}^{m-1}\{i_q\}} e^{\theta_x}} \left(1 - \frac{e^{\theta_i}+e^{\theta_j}}{\sum_{x \in \mathcal{X}\setminus\{i_1,\ldots,i_{k-2}\}} e^{\theta_x}}\right)$$

$$\geq \sum_{(i_1,\ldots,i_{k-2}) \in \mathcal{X}\setminus\{i,j\}} \prod_{m=1}^{k-2} \frac{e^{\theta_{i_m}}}{\sum_{x \in \mathcal{X}\setminus \cup_{q=1}^{m-1}\{i_q\}} e^{\theta_x}} \left(1 - \frac{e^{\theta_i}+e^{\theta_j}}{e^{\theta_i}+e^{\theta_j}+(n-k)e^{-B}}\right)$$

$$= \sum_{(i_1,\ldots,i_{k-3}) \in \mathcal{X}\setminus\{i,j\}} \prod_{m=1}^{k-3} \frac{e^{\theta_{i_m}}}{\sum_{x \in \mathcal{X}\setminus \cup_{q=1}^{m-1}\{i_q\}} e^{\theta_x}} \left(1 - \frac{e^{\theta_i}+e^{\theta_j}}{\sum_{x \in \mathcal{X}\setminus\{i_1,\ldots,i_{k-3}\}} e^{\theta_x}}\right)$$

$$\times \left(1 - \frac{e^{\theta_i}+e^{\theta_j}}{e^{\theta_i}+e^{\theta_j}+(n-k)e^{-B}}\right)$$

$$\geq \sum_{(i_1,\ldots,i_{k-3}) \in \mathcal{X}\setminus\{i,j\}} \prod_{m=1}^{k-3} \frac{e^{\theta_{i_m}}}{\sum_{x \in \mathcal{X}\setminus \cup_{q=1}^{m-1}\{i_q\}} e^{\theta_x}} \left(1 - \frac{e^{\theta_i}+e^{\theta_j}}{e^{\theta_i}+e^{\theta_j}+(n-k-1)e^{-B}}\right)$$

$$\times \left(1 - \frac{e^{\theta_i}+e^{\theta_j}}{e^{\theta_i}+e^{\theta_j}+(n-k)e^{-B}}\right)$$

$$...$$

$$\geq \prod_{m=1}^{k-1} \left(1 - \frac{e^{\theta_i}+e^{\theta_j}}{e^{\theta_i}+e^{\theta_j}+(n-m-1)e^{-B}}\right)$$

$$\geq \prod_{m=1}^{k-1} \left(1 - \frac{2e^B}{2e^B+(n-m-1)e^{-B}}\right).$$

The first line restates the equation, the second lower bounds the final term by decreasing the logits of all variables but $\theta_i$ and $\theta_j$ to $-B$, their smallest possible value. The lower bound is then restated in the third line, where an explicit specification of the probability of choice $k-2$ is stated implicitly as the probability of any choice in the available universe but $i$ and $j$. We may perform this step after the lower bound, but not before, because the probability of choice $k-1$ in the lower bound is unaffected by the choice made in $k-2$, whereas it would be affected in the original bound. In the fourth line, we lower bound the $k-2$ choice probability in a similar manner to the second line, and repeat the restate-bound procedure over and over until we arrive at the final inequality, stated in the second to last line. This expression is again lower bounded in the last line by raising the utility of items $i$ and $j$ to make the lower bound independent of $i$ and $j$.

Next, we have,

$$P(X_{ij} \geq k) \geq \prod_{m=1}^{k-1} \left(1 - \frac{2e^B}{2e^B+(n-m-1)e^{-B}}\right)$$

$$\geq \left(1 - \frac{2e^B}{2e^B+(n-k)e^{-B}}\right)^{k-1}$$

$$\geq 1 - \frac{2(k-1)e^B}{2e^B+(n-k)e^{-B}}$$

$$\geq 1 - \frac{2ke^B}{2e^B+(n-k)e^{-B}},$$

and so for $\delta \in [0,1]$,

$$P(X_{ij} \geq \delta n) \geq 1 - \frac{2e^{2B}\delta n}{2e^{2B}+(1-\delta)ne^{-B}}$$

$$= 1 - \frac{\delta n}{1+(1-\delta)n\frac{e^{-3B}}{2}}$$

$$\geq 1 - \frac{2e^{3B}\delta}{(1-\delta)}.$$

Now, setting

$$\delta = \frac{1-c}{2e^{3B}+1-c},$$

we have that

$$P\left(X_{ij} \geq \frac{1-c}{2e^{3B}+1-c}n\right) \geq c.$$

This expression claims that $X_{ij}$ is at least linear in $n$ with however large a probability we desire. We can use these expressions to also lower bound the expectation:

$$\mathbb{E}[X_{ij}] = 1 + \sum_{k=2}^{n-1} P(X_{ij} \geq k)$$

$$\geq 1 + \sum_{k=2}^{\frac{n}{1+2e^{3B}}} \left(1 - \frac{2e^{3B}k}{n-k}\right)$$

$$= 1 + \left(\frac{n}{1+2e^{3B}} - 1\right) - 2e^{3B} \sum_{k=2}^{\frac{n}{1+2e^{3B}}} \frac{k}{n-k}$$

$$\geq 1 + \left(\frac{n}{1+2e^{3B}} - 1\right) - 2e^{3B} \sum_{k=2}^{\frac{n}{1+2e^{3B}}} \frac{k}{n - \frac{n}{1+2e^{3B}}}$$

$$= 1 + \left(\frac{n}{1+2e^{3B}} - 1\right) - 2e^{3B}(1+2e^{3B}) \sum_{k=2}^{\frac{n}{1+2e^{3B}}} \frac{k}{(1+2e^{3B})n - n}$$

$$= 1 + \left(\frac{n}{1+2e^{3B}} - 1\right) - \frac{1+2e^{3B}}{n} \sum_{k=2}^{\frac{n}{1+2e^{3B}}} k$$

$$= 1 + \left(\frac{n}{1+2e^{3B}} - 1\right) - \frac{1+2e^{3B}}{n} \frac{1}{2}\left(\frac{n}{1+2e^{3B}}\right)\left(\frac{n}{1+2e^{3B}} + 1\right) + 1$$

$$= \frac{1}{2}\left(\frac{n}{1+2e^{3B}} + 1\right),$$

where the first expression follows from the substitution of the tail bound only for the regime where it is non-zero (ignoring, for simplicity, the matter of the ceiling operators), and using zero otherwise, the next follows from lower bounding the second expression with the lowest value achieved in the sum. The remaining steps perform the necessary algebra to arrive at the final expression, which is clearly affine in $n$.

Now, recall that $\bar{X}_{ij}$ is the mean of $\ell$ independent variables $X_{ij}$. Certainly, the expectation of $\bar{X}_{ij}$ is the same as that of $X_{ij}$. Since $X_{ij}$ is always bounded between 1 and $n-1$, we may use the one-sided Hoeffding's inequality to make the following claim, for any $i, j$:

$$P(\mathbb{E}[X_{ij}] - \bar{X}_{ij} \geq \delta n) \leq \exp\left(-\frac{\ell^2\delta^2 n^2}{\ell(n-2)^2}\right) \leq \exp(-\ell\delta^2).$$

That is, for any $i, j$, with probability at least $1 - \exp(-\ell\delta^2)$ we have that $\mathbb{E}[X_{ij}] - \bar{X}_{ij} \leq \delta n$. Using our lower bound from before, we have that

$$\frac{1}{2}\left(\frac{n}{1+2e^{3B}} + 1\right) - \delta n \leq \bar{X}_{ij}.$$

A strong upper bound on the failure probability of $\bar{X}_{ij}$ attaining the left-hand-side value allows us to easily union bound the failure probability of $\min_{ij} \bar{X}_{ij}$ attaining the same left-hand-side value. Namely,

$$\min_{ij} \bar{X}_{ij} < \frac{1}{2}\left(\frac{n}{1+2e^{3B}} + 1\right) - \delta n$$

if any $\bar{X}_{ij}$ is less than the right hand side. Thus,

$$\mathbb{P}\Big[\min_{ij} \bar{X}_{ij} < \frac{1}{2}\Big(\frac{n}{1 + 2e^{3B}} + 1\Big) - \delta n\Big] \leq \sum_{ij} \mathbb{P}\Big[\bar{X}_{ij} < \frac{1}{2}\Big(\frac{n}{1 + 2e^{3B}} + 1\Big) - \delta n\Big] \leq n^2 \exp(-\ell\delta^2),$$

where the first inequality is the union bound, and the second inequality applies the preceding bound on the failure probability of any $\bar{X}_{ij}$ and additionally upper bounds as $\binom{n}{2}$ by $n^2$.

Now, Setting $\delta$ to $\frac{1}{4}\frac{1}{1+2e^{3B}}$, we may conclude the following: with probability at least

$$1 - n^2 \exp\Big(-\ell \frac{1}{16}\frac{1}{(1+2e^{3B})^2}\Big), \tag{18}$$

we have that

$$\frac{1}{4}\Big(\frac{n}{1 + 2e^{3B}} + 2\Big) \leq \min_{ij} \bar{X}_{ij}.$$

Returning all the way to Equation (17), we have that:

$$\lambda_2(L) \geq \frac{n}{n-1}\min_{ij} \bar{X}_{ij} \geq \frac{n}{n-1}\frac{1}{4}\Big(\frac{n}{1 + 2e^{3B}} + 2\Big).$$

Defining $\alpha_B$ as

$$\alpha_B := \frac{1}{4(1 + 2e^{3B})},$$

which simplifies to:

$$\lambda_2(L) \geq \frac{n}{n-1}\Big(\alpha_B n + \frac{1}{2}\Big) \geq \alpha_B n, \tag{19}$$

where we lower bound $\frac{n}{n-1}$ by 1 and drop the $1/2$.

Rewriting the probability of this inequality (from (18)) in terms of $\alpha_B$, we have that Equation (19) occurs with probability

$$1 - n^2 \exp(-\ell\alpha_B), \tag{20}$$

completing the proof. □

**Lemma 5.** *Let $\sigma_1, \ldots, \sigma_\ell$ be drawn iid CRS($u^\star$), the full CRS model with true parameter $u^\star \in \Theta_B = \{u \in \mathbb{R}^{n(n-1)} : u = [u_1^T, ..., u_n^T]^T; u_i \in \mathbb{R}^{n-1}, \|u_i\|_1 \leq B, \forall i; \mathbf{1}^T u = 0\}$. Let $\lambda_2(X^T X)$ be the second smallest eigenvalue of the scaled (random) design matrix $X$ obtained from $\ell$ samples of CRS($u^\star$). Then*

$$\frac{\mu_B}{n^3(n-1)} < \lambda_2(X^T X),$$

*with probability at least $1 - n\exp(-\frac{\ell}{8ne^{2B}})$, where $\mu_B = 1/(4e^{2B})$.*

**Proof.** Define $L = X^T X$, denote by $\mathcal{D}$ the collection of $\ell(n-1)$ choices constructed from the $\ell$ rankings by repeated selection, and let $C_{\mathcal{D}}$ be the set of choice sets in $\mathcal{D}$. We also use $X(\mathcal{D})$ to refer to $X$. The CDM is identifiable (and so $\lambda_2(L) > 0$) the moment $C_{\mathcal{D}}$ contains the set $\mathcal{X}$ and $\mathcal{X} \setminus i, \forall i$. We can see this observation because when we have all $\mathcal{X} \setminus i$, we have all sets of size $n$ and $n - 1$, and thus can invoke Theorem 4 of [48] to say that $\text{rank}(X(\mathcal{D})) = n(n-1) - 1$ and thus $\text{rank}(L) \leq \text{rank}(X(\mathcal{D})^T X(\mathcal{D})) = \text{rank}(X(\mathcal{D})) = n(n-1) - 1$, so $\lambda_2(L) > 0$.

The universe set $\mathcal{X}$ always appears with every ranking, so we could aim to lower bound the probability that each set of size $n - 1$ appears in $C_{\mathcal{D}}$ at least once in order to get identifiability. We can however, do more if each set of size $n - 1$ appears in $C_{\mathcal{D}}$ at least $r$ times.

Suppose, for instance, that all $\mathcal{X} \setminus i$ are in $C_{\mathcal{D}}$. Consider just the choices corresponding to the $n$ unique sets of size $n - 1$, as well as the $n$ universe sets that came along with the sets of $n - 1$ in the same ranking, and refer to this collection of choices by the dataset $\tilde{\mathcal{D}}$. For this dataset, the corresponding $L$ matrix (which we will label $L_{\tilde{\mathcal{D}}}$) is, using the notation of [48],

$$L_{\tilde{\mathcal{D}}} = \frac{1}{2n}\sum_{i=1}^{2n} E_{i,k_i}^T\Big(I - \frac{1}{k_i}\mathbf{1}\mathbf{1}^T\Big)E_{i,k_i}.$$

where the matrices $E_{i,k_i} \in \{0,1\}^{n(n-1) \times k_i}$ depend only on the sets (not the hoices). We now apply Lemma 6, noting that the $L_{\tilde{\mathcal{D}}}$ in the statement of the Lemma is identical to the one here:

$$\delta_n := \frac{1}{4n^3} < \lambda_2(L_{\tilde{\mathcal{D}}}). \tag{21}$$

Now, look back at $\mathcal{D}$, which has $\ell$ rankings, and the $n-1$ choices that come from each. $L$ is thus:

$$L = \frac{1}{\ell(n-1)} \sum_{i=1}^{\ell(n-1)} E_{i,k_i}(I - \frac{1}{k_i}\mathbf{1}\mathbf{1}^T)E_{i,k_i}^T$$

Since $\mathcal{D}$ contains the choices in $\tilde{\mathcal{D}}$, and since each term in the sum of $L$ is PSD,

$$\frac{2n}{\ell(n-1)}\delta_n \le \lambda_2(L),$$

as a result of properties of sums of PSD matrices [11][See pg. 128, Corollary (4.2)]. In fact, if $\mathcal{D}$ contains the choices $\tilde{\mathcal{D}}$ in $r$ copies, then we can say

$$\frac{2nr}{\ell(n-1)}\delta_n \le \lambda_2(L).$$

With this understanding, we may now proceed to study the probability that each set of size $n-1$ appears in $C_{\mathcal{D}}$ at least $r$ times. Let $\sigma \sim CRS(u^\star)$. The subsets $\mathcal{X} \setminus i$ appear in the repeated selection decomposition of a ranking if $\sigma^{-1}(i) = 1$, that is, if $i$ is ranked first, which happens with probability

$$P(i|\mathcal{X}; u^*) \propto \exp(\sum_{j \in \mathcal{X} \setminus i} u_{ij}^*).$$

Noting that $||u_i^*||_1 \le B$, it follows that $\sum_{j \in \mathcal{X} \setminus i} u_{ij}^* \ge -B$ and that for every $i' \ne i$, $\sum_{j \in \mathcal{X} \setminus i'} u_{i'j}^* \le B$. It follows that

$$P(i|\mathcal{X}; u^*) \ge \frac{e^{-B}}{e^{-B} + (n-1)e^B} \ge \frac{1}{ne^{2B}}.$$

Consider now $\mathcal{D}$, which are $\ell$ rankings from the model, $\sigma_1, \ldots, \sigma_{\ell_0} \sim CRS(u^*)$. The probability that $\mathcal{X} \setminus i \in C_{\mathcal{D}}$ at most $r_i$ times given $\ell$ trials is the CDF of a Binomial distribution with success probability $P(i|\mathcal{X}; u^*)$. We may use a Chernoff bound to upper bound the CDF [2]:

$$P(\mathcal{X} \setminus i \in C_{\mathcal{D}} \text{ at most } r_i \text{ times}) \le \exp\left(-\ell D\left(\frac{r_i}{\ell}||P(i|\mathcal{X}; u^*)\right)\right)$$
$$\le \exp\left(-\ell D\left(0.5 P(i|\mathcal{X}; u^*)||P(i|\mathcal{X}; u^*)\right)\right)$$
$$\le \exp\left(-\ell 0.125 P(i|\mathcal{X}; u^*)\right)$$
$$\le \exp\left(-\frac{\ell}{8ne^{2B}}\right),$$

where the second inequality follows by setting $r_i = .5 P(i|\mathcal{X}; u^*)\ell$, the third from the fact that $D(.5p||p) \ge .125p$, and the last from lower bounding $P(i|\mathcal{X}; u^*)$. For each $i \in \mathcal{X}$, let $A_i$ be the event that $\mathcal{X} \setminus i \in C_{\mathcal{D}}$ at most $r_i$ times. Then all $\mathcal{X} \setminus i$ are in $C_{\mathcal{D}}$ at least $r = \min_i r_i$ times whenever we are *not* in $\cup_{i \in \mathcal{X}} A_i^C$. With a union bound and the previous bound we have

$$P(\cup_{i \in \mathcal{X}} A_i^C) \le \sum_i P(A_i^C)$$
$$\le n \exp\left(-\frac{\ell}{8ne^{2B}}\right).$$

With this result, we can see that with probability at least $1 - n \exp(-\frac{\ell}{8ne^{2B}})$,

$$\lambda_2(L) \ge \frac{2nr}{\ell(n-1)}\delta_n \ge \frac{2n}{\ell(n-1)}\frac{\ell}{2ne^{2B}}\delta_n \ge \frac{\delta_n}{(n-1)e^{2B}} = \frac{1}{4n^3(n-1)e^{2B}},$$

where we substitute the value of $\delta_n$ from (21) in the last equality, which completes the proof. $\square$

**Lemma 6.** *Consider the matrix*

$$L_{\tilde{\mathcal{D}}} = \frac{1}{2} E_{\mathcal{X}}^T (I - \frac{1}{n} \mathbf{1}\mathbf{1}^T) E_{\mathcal{X}} + \frac{1}{2n} \sum_{x \in \mathcal{X}} E_{\mathcal{X} \setminus x}^T (I - \frac{1}{n-1} \mathbf{1}\mathbf{1}^T) E_{\mathcal{X} \setminus x}.$$

*where* $\mathbf{1}$ *is the all ones vector and the matrices* $E_C \in \{0,1\}^{n(n-1) \times |C|}$, *for any subset* $C \subseteq \mathcal{X}$ *is defined equivalently to* $E_{i,k_i}$ *in Lemma 7. We have,*

$$\frac{1}{4n^3} < \lambda_2(L_{\tilde{\mathcal{D}}}).$$

**Proof.** The matrix $E_C$ selects specific elements of the CDM parameter vector $u$ corresponding to ordered pairs within the set $C$. $E_C$ has a column for every item $y \in C$ (and hence $|C|$ columns), and the column corresponding to item $y \in C$ has a one at the position of each $u_{yz}$ for $z \in C \setminus y$, and zero otherwise. Thus, the vector-matrix product $u^T E_C = [\sum_{z \in C \setminus y_1} u_{y_1 z}, \sum_{z \in C \setminus y_2} u_{y_2 z}, \cdots \sum_{z \in C \setminus y_{|C|}} u_{y_{k_j} z}] \in \mathbb{R}^{1 \times |C|}$. Throughout the proof we will operate on the matrix $L = 2n L_{\tilde{\mathcal{D}}}$, and then carry our results over to $L_{\tilde{\mathcal{D}}}$ at the end.

Denote by $g_y$ the column of $E_{\mathcal{X}}$ corresponding to item $y$. Additionally, define $h_y \in \{0,1\}^{n(n-1)}$ as the vector that has a one at the position of each $u_{yz}$ for $z \in \mathcal{X} \setminus y$ and $u_{zy}$ for $z \in \mathcal{X} \setminus y$, and zero otherwise. With these new definitions, we may perform some manipulations to rewrite $L$ as,

$$L = (2n - 3) \sum_{x \in \mathcal{X}} g_x g_x^T - \left(1 + \frac{n-4}{n-1}\right) \mathbf{1}\mathbf{1}^T + I - \frac{1}{n-1} \sum_{x \in \mathcal{X}} h_x h_x^T.$$

Now, consider the subspace $V$ spanned by the columns $g_x$ and $h_x$, $\forall x \in \mathcal{X}$ as well as the all ones vector $\mathbf{1}$. Evidently, for any vector $w \in \text{Null}(V)$, the kernel of the subspace $V$, we have,

$$Lw = (2n-3) \sum_{x \in \mathcal{X}} g_x g_x^T w - \left(1 + \frac{n-4}{n-1}\right) \mathbf{1}\mathbf{1}^T w + Iw - \frac{1}{n-1} \sum_{x \in \mathcal{X}} h_x h_x^T w = Iw = w,$$

where the second equality follows because $w$ is orthogonal to all the vectors it is is dotting, by definition of $w$ being orthogonal to the subspace $V$. Thus, every vector $w \in \text{Null}(V)$ is an eigenvector with eigenvalue 1. Moreover, any eigenvector $v$ with eigenvalue *not* equaling 1 must belong in $V$. If we can find a collection of eigenvectors that span the subspace $V$, then the second smallest eigenvalue must either be an eigenvalue corresponding to one of those eigenvectors, or be 1, corresponding to any vector in $\text{Null}(V)$. We now proceed to construct a collection of eigenvectors that span the subspace $V$.

Evidently, $\mathbf{1}$ is an eigenvector with eigenvalue 0, since $L\mathbf{1} = 0$. Next, define

$$v_x^b = [-b - (n-2)]g_x + b(h_x - g_x) + (\mathbf{1} - h_x).$$

for every $x \in \mathcal{X}$ and for any scalar $b$. We note that each of the three component vectors in the definition of $v_x^b$ ($g_x$, $h_x - g_x$, and $(\mathbf{1} - h_x)$) all have non-zero entries in only the coordinates for which the other two vectors have zeros. Consider $v_x^b$ for two unique $b$ values, $b_1 \neq b_2$, and for all $x \in \mathcal{X}$. We can show that the $v_x^{b_1}$ and $v_x^{b_2}$ collection of vectors, along with the $\mathbf{1}$ vector, span the subspace $V$.

We start by noticing that $v_x^{b_1} - \mathbf{1} = [-2b_1 - (n-2)]g_x + (b_1 - 1)h_x$ and $v_x^{b_2} - \mathbf{1} = [-2b_2 - (n-2)]g_x + (b_2 - 1)h_x$. And so we can perform linear manipulations to find $g_x$ and $h_x$, $\forall x \in \mathcal{X}$. Thus, the $v_x^{b_1}$ and $v_x^{b_2}$ collection of vectors, along with the $\mathbf{1}$ vector, span the subspace $V$, since all the constituent vectors of $V$ are within the range of those vectors.

Now, we proceed to show that $v_x^b$ is an eigenvector for two unique $b$ values for all $x \in \mathcal{X}$. We have,

$$Lv_x^b = \left[ -n(n-1)b - n(n-1)(n-2) - (n-2)^2 b - (n-2)^3 + \frac{(n-2)^2}{n-1} \right] g_x +$$

$$\left[ (2n-2)b + \frac{2n^3 - 8n^2 + 9n - 2}{n-1} \right] (h_x - g_x) +$$

$$\left[ (2n-3)b + \frac{2n^3 - 9n^2 + 12n - 3}{n-1} \right] (\mathbf{1} - h_x).$$

In order for $v_x^b$ to be an eigenvector for some value $b$, it must by definition satisfy $Lv_x^b = \lambda_b v_x^b$, where $\lambda_b$ is the corresponding eigenvalue. Since the vector components $g_x$, $h_x - g_x$, and $(\mathbf{1} - h_x)$ have non-zero values in unique coordinates, $Lv_x^b = \lambda_b v_x^b$ can be stated as equalities on the respective coefficients of $v_x^b$ and $Lv_x^b$:

$$\left[ -n(n-1)b - n(n-1)(n-2) - (n-2)^2 b - (n-2)^3 + \frac{(n-2)^2}{n-1} \right] = \lambda_b[-b - (n-2)],$$

$$\left[ (2n-2)b + \frac{2n^3 - 8n^2 + 9n - 2}{n-1} \right] = \lambda_b b,$$

$$\left[ (2n-3)b + \frac{2n^3 - 9n^2 + 12n - 3}{n-1} \right] = \lambda_b.$$

A simple calculation reveals that satisfying the latter two constraints satisfies the former. We can then solve for $b$ by substituting the latter two constraints into one equation,

$$\left[ (2n-2)b + \frac{2n^3 - 8n^2 + 9n - 2}{n-1} \right] = \left[ (2n-3)b + \frac{2n^3 - 9n^2 + 12n - 3}{n-1} \right] b,$$

and can then substitute solutions for $b$ into the expression for $\lambda_b$ ($v_x^b$) to find the corresponding eigenvalue (eigenvector). Rearranging terms, we find that solving for $b$ amounts to finding the root of a quadratic:

$$(2n^2 - 5n + 3)b^2 + (2n^3 - 11n^2 + 16n - 5)b - (2n^3 - 8n^2 + 9n - 2) = 0.$$

Applying the quadratic formula yields two solutions for $b$:

$$b = \frac{-2n^3 + 11n^2 - 16n + 5 \pm \sqrt{4n^6 - 28n^5 + 81n^4 - 116n^3 + 74n^2 - 12n + 1}}{4n^2 - 10n + 6}.$$

A root finding algorithm demonstrates that the discriminant is strictly positive for all $n \geq 2$, and so there are always two unique solutions, $b_1$ and $b_2$. Two unique solutions then means we have two sets of eigenvectors $v_x^{b_1}$ and $v_x^{b_2}$, $\forall x \in \mathcal{X}$. As shown previously, two sets of vectors $v_x^{b_1}$ and $v_x^{b_2}$ along with $\mathbf{1}$ spans the subspace of $V$. Thus, we have eigenvectors that span the subspace $V$, and know that all vectors in $\mathrm{Null}(V)$ are eigenvectors with eigenvalue $1$.

Now, substituting the solutions for $b$ into the expression for $\lambda_b$ yields the corresponding pair of eigenvalues:

$$\lambda_b = \frac{2n^3 - 7n^2 + 8n - 1 \pm \sqrt{4n^6 - 28n^5 + 81n^4 - 116n^3 + 74n^2 - 12n + 1}}{2(n-1)}.$$

Ostensibly, the "minus" solution above corresponds to the smaller of the two eigenvalues. Denote the smaller by just $\lambda$. Should this smaller eigenvalue be less than $1$ and greater than $0$, it is the second smallest eigenvalue. If it fails to We now show this is the case. Set $\alpha := (2n^3 - 7n^2 + 8n - 1)^2$ and $\beta := 4n^6 - 28n^5 + 81n^4 - 116n^3 + 74n^2 - 12n + 1$. We then have that

$$\lambda = \frac{\sqrt{\alpha} - \sqrt{\alpha - 4n^2 + 4n}}{2(n-1)} = \sqrt{4n^2 - 4n} \frac{\sqrt{\frac{\alpha}{4n^2 - 4n}} - \sqrt{\frac{\alpha}{4n^2 - 4n} - 1}}{2(n-1)}$$

. Using the well known inequality

$$\frac{1}{2\sqrt{x}} < \sqrt{x} - \sqrt{x-1}, \quad \forall x \geq 1,$$

we have a lower bound,

$$\lambda > \sqrt{4n^2 - 4n} \frac{1}{4(n-1)\sqrt{\frac{\alpha}{4n^2 - 4n}}} = \frac{n}{\sqrt{\alpha}} = \frac{n}{2n^3 - 7n^2 + 8n - 1} \geq \frac{n}{2n^3} = \frac{1}{2n^2} > 0.$$

For the upper bound, we have that

$$\lambda = \frac{\sqrt{\beta + 4n^2 - 4n} - \sqrt{\beta}}{2(n-1)} = \sqrt{4n^2 - 4n} \frac{\sqrt{\frac{\beta}{4n^2 - 4n} + 1} - \sqrt{\frac{\beta}{4n^2 - 4n}}}{2(n-1)}.$$

Using the well known inequality

$$\frac{1}{2\sqrt{x}} > \sqrt{x+1} - \sqrt{x}, \ \ \forall x \geq 1,$$

we have,

$$\lambda < \sqrt{4n^2 - 4n}\frac{1}{4(n-1)\sqrt{\frac{\beta}{4n^2-4n}}} = \frac{n}{\sqrt{\beta}} \leq \frac{n}{(n-1)^3}.$$

This upper bound is less than 1 so long as $n \geq 3$. Because $\lambda$ is less than 1, we know that 1 is not the second smallest eigenvalue (corresponding to any vector in the subspace $\text{Null}(V)$). Because $\lambda$ is greater than 0, and the smaller of the two values $\lambda_{b_1}$ and $\lambda_{b_1}$, it is indeed the second smallest eigenvalue of $L$. Furthermore, we have the lower bound from above, $\lambda > 1/2n^2$.

Since, $L_{\tilde{\mathcal{D}}} = \frac{1}{2n}L$, we can conclude that $\lambda_2(L_{\tilde{\mathcal{D}}}) = \frac{1}{2n}\lambda > \frac{1}{4n^3}$, which proves the theorem statement.

$\square$

**Lemma 7.** *Let $u^\star$ denote the true CDM model from which choice data $\mathcal{D}$ with $m$ choices is drawn. Let $\hat{u}_{MLE}$ denote the maximum likelihood solution. For any $u^\star \in \mathcal{U}_B = \{u \in \mathbb{R}^{n(n-1)} : u = [u_1^T, ..., u_n^T]^T; u_i \in \mathbb{R}^{n-1}, \|u_i\|_1 \leq B, \forall i; \boldsymbol{1}^T u = 0\}$, and $t > 1$,*

$$\mathbb{P}\Big[\|\hat{u}_{MLE}(\mathcal{D}) - u^\star\|_2^2 \geq c_{B,k_{max}}\frac{ctn(n-1)}{m\lambda_2(L)}\Big] \leq e^{-t},$$

*where $k_{max}$ is the maximum choice set size in $\mathcal{D}$, $c_{B,k_{max}}$ is a constant that depends on $B$ and $k_{max}$, and $\lambda_2(L)$ the spectrum of $L = X(\mathcal{D})^T X(\mathcal{D})$ with scaled design matrix $X(\mathcal{D})$. For the expected risk,*

$$\mathbb{E}\big[\|\hat{u}_{MLE}(\mathcal{D}) - u^\star\|_2^2\big] \leq c'_{B,k_{max}}\frac{n(n-1)}{m\lambda_2(L)},$$

*where the expectation is taken over the dataset $\mathcal{D}$ generated by the choice model and $c'_{B,k_{max}}$ is again a constant that depends on $B$ and $k_{max}$.*

**Proof.** Our proof is very similar to the proof of Theorem 1. Much like that proof, we will first introduction notation for analyzing the risk, and then proceed to first give a proof of the expected risk bound. We then carry the technology of that proof forward to give a proof of the tail bound statement. The great similarity to the proof of Theorem 1 demands that we repeat some of the same arguments here – in lieu of doing that, we jump to conclusions and refer the reader to the respective sections of Theorem 1. The resulting risk bound portion is a nearly exact replica of the risk bound in [48] (see Theorem 1), with the main difference being that our statement here employs a more restrictive assumption on $\mathcal{U}_B$ than its counterpart in [48], based on an infinity-norm vs. the 1-norm above. We constrain $\mathcal{U}_B$ so we may apply Lemma 1 to efficiently bound a $\beta_{k_{max}}$ term that arises in the proof. Unpacking the proof for the risk bound additionally yields the right tools to prove the tail bound result above, an entirely novel contribution of our work. For both the tail and risk bound sections, the notation we use for the CDM model often overloads the notation used in the MNL model's proof. This overloading is intentional, and is meant to convey the high degree of similarity between the two models, and the proof techniques used ot provide guarantees for them.

We are given some true CDM model with parameters $u^\star \in \mathcal{U}_B$, and for each datapoint $j \in [m]$ we have the probability of choosing item $x$ from set $C_j$ as

$$\mathbb{P}(y_j = x | u^\star, C_j) = \frac{\exp(\sum_{z \in C_j \setminus x} u_{xz}^\star)}{\sum_{y \in C_j} \exp(\sum_{z \in C_j \setminus y} u_{yz}^\star))}.$$

**Notation.** We now introduce notation that will let us represent the above expression in a more compact manner. In this proof, we will use $d = n(n-1)$ to refer to the CDM parameter space. Because our datasets involve choice sets of multiple sizes, we use $k_j \in [k_{min}, k_{max}]$ to denote the choice set size for datapoint $j$, $|C_j|$. Extending a similar concept in [49] to the multiple set sizes, and the more complex structure of the CDM, we then define matrices $E_{j,k_j} \in \mathbb{R}^{d \times k_j}, \forall j \in [m]$ as follows: $E_{j,k_j}$ has a column for every item $y \in C_j$ (and hence $k_j$ columns), and the column corresponding to item $y \in C_j$ has a one at the position of each $u_{yz}$ for $z \in C_j \setminus y$, and zero otherwise. This construction

allows us to write the familiar expressions $\sum_{z \in C_j \setminus y} u_{yz}$, for each $y$, simply as a single vector-matrix product $u^T E_{j,k_j} = [\sum_{z \in C_j \setminus y_1} u_{y_1 z}, \sum_{z \in C_j \setminus y_2} u_{y_2 z}, \ldots \sum_{z \in C_j \setminus y_{k_j}} u_{y_{k_j} z}] \in \mathbb{R}^{1 \times k_j}$.

Next, we define a collection of functions $F_k : \mathbb{R}^k \mapsto [0, 1], \forall k \in [k_{\min}, k_{\max}]$ as

$$F_k([x_1, x_2, \ldots, x_k]) = \frac{\exp(x_1)}{\sum_{l=1}^k \exp(x_l)},$$

where the numerator always corresponds to the first entry of the input. These functions $F_k$ have several properties that will become useful later in the proof. First, it is easy to verify that all $F_k$ are shift-invariant, that is, $F_k(x) = F_k(x + c\mathbf{1})$, for any scalar $c$. The purpose of introducing $F_k$ is to write CDM probabilities compactly. Since the inputs $x_i$ are sums of $k-1$ values of the $u$ vector, the inputs $x \in [-B, B]^n$.

We may thus apply Lemma 1, and obtain that that $\mathbf{1} \in \text{null}(\nabla^2(-\log(F_k(x))))$ and that

$$\nabla^2(-\log(F_k(x))) \succeq H_k = \beta_k(I - \frac{1}{k}\mathbf{1}\mathbf{1}^T), \tag{22}$$

where

$$\beta_k := \frac{1}{k \exp(2B)}. \tag{23}$$

That is, $F_k$ are strongly log-concave with a null space *only* in the direction of $\mathbf{1}$, since $\nabla^2(-\log(F_k(x))) \succeq H_k$ for some $H_k \in \mathbb{R}^{k \times k}, \lambda_2(H_k) > 0$.

As a final notational addition, in the same manner as [49] but accounting for multiple set sizes, we define $k$ permutation matrices $R_{1,k}, \ldots, R_{k,k} \in \mathbb{R}^{k,k}, \forall k \in [k_{\min}, k_{\max}]$, representing $k$ cyclic shifts in a fixed direction. Specifically, given some vector $x \in \mathbb{R}^k$, $y = x^T R_{l,k}$ is simply $x^T$ cycled (say, clockwise) so $y_1 = x_l, y_i = x_{(l+i-1)\%k}$, where $\%$ is the modulo operator. That is, these matrices allow for the cycling of the entries of row vector $v \in \mathbb{R}^{1 \times k}$ so that any entry can become the first entry of the vector, for any of the relevant $k$. This construction allows us to represent any choice made from the choice set $C_j$ as the first element of the vector $x$ that is input to $F$, thereby placing it in the numerator.

**First, an expected risk bound.** Given the notation introduced above, we can now state the probability of choosing the item $x$ from set $C_j$ compactly as:

$$\mathbb{P}(y_j = x|u^\star, C_j) = \mathbb{P}(y_j = x|u^\star, k_j, E_{j,k_j}) = F_{k_j}(u^{\star T} E_{j,k_j} R_{x,k_j}).$$

We can then rewrite the CDM likelihood as

$$\sup_{u \in \mathcal{U}_B} \prod_{(x_j, k_j, E_{j,k_j}) \in \mathcal{D}} F_{k_j}(u^T E_{j,k_j} R_{x_j, k_j}),$$

and the scaled negative log-likelihood as

$$\ell(u) = -\frac{1}{m} \sum_{(x_j, k_j, E_{j,k_j}) \in \mathcal{D}} \log(F_{k_j}(u^T E_{j,k_j} R_{x_j, k_j})) = -\frac{1}{m} \sum_{j=1}^m \sum_{i=1}^{k_j} \mathbf{1}[y_j = i] \log(F_{k_j}(u^T E_{j,k_j} R_{i,k_j})).$$

Thus,

$$\hat{u}_{\text{MLE}} = \arg\max_{u \in \mathcal{U}_B} \ell(u).$$

At this point, it should be clear to the reader that the problem formulation is almost exactly the same as that of Theorem 1, the only difference being that $u$ belongs in a higher dimensional space than $\theta$, and that the respective proofs' $E_{j,k_j}$ is defined differently. Due to $\mathcal{U}_B$'s definition restricting the $\ell_1$ norm of certain entries of $u$, we see that the inputs $u^T E_{j,k_j} R_{x_j, k_j}$ to the functions $F_{k_j}$ live within $[-B, B]^{k_j}$, much like the restrictions on $\Theta_B$ also resulted in $\theta^T E_{j,k_j} R_{x_j, k_j} \in [-B, B]^{k_j}$. The similarity of the problems allow us to port over certain steps used in the proof without additional justification.

To begin, we have the gradient of the negative log-likelihood as

$$\nabla \ell(u) = -\frac{1}{m} \sum_{j=1}^{m} \sum_{i=1}^{k_j} \mathbf{1}[y_j = i] E_{j,k_j} R_{i,k_j} \nabla \log(F_{k_j}(u^T E_{j,k_j} R_{i,k_j})), \tag{24}$$

and the Hessian as

$$\nabla^2 \ell(u) = -\frac{1}{m} \sum_{j=1}^{m} \sum_{i=1}^{k_j} \mathbf{1}[y_j = i] E_{j,k_j} R_{i,k_j} \nabla^2 \log(F_{k_j}(u^T E_{j,k_j} R_{i,k_j})) R_{i,k_j}^T E_{j,k_j}^T.$$

Proceeding identically as Theorem 1, we have, for any vector $z \in \mathbb{R}^d$,

$$z^T \nabla^2 \ell(u) z \geq \beta_{k_{\max}} \frac{1}{m} \sum_{j=1}^{m} z^T E_{j,k_j} (I - \frac{1}{k_j} \mathbf{1}\mathbf{1}^T) E_{j,k_j}^T z,$$

where we have followed the same steps taken in Theorem 1 to bound $z^T \nabla^2 \ell(\theta) z$. Now, defining the matrix $L$ as

$$L = \frac{1}{m} \sum_{j=1}^{m} E_{j,k_j} (I - \frac{1}{k_j} \mathbf{1}\mathbf{1}^T) E_{j,k_j}^T,$$

we first note a few properties of $L$. First, it is easy to verify that, like in Theorem 1, $L\mathbf{1} = 0$, and hence span$(\mathbf{1}) \subseteq$ null$(L)$. It is also straightforward to see that $L$ is **not** a Laplacian, unlike the MNL case, since every non pair edge creates positive entries in the off diagaonal. Moreover, we follow an argument in [48] to show that $\lambda_2(L) > 0$, that is, null$(L) \subseteq$ span$(\mathbf{1})$. Consider the matrix $G(\mathcal{D})$, the design matrix of the CDM for the given dataset $\mathcal{D}$ described in detail in Theorem 4 of [48]. Define a matrix $X(\mathcal{D}) = \mathbf{C}_{\mathcal{D}}^{-1} G(\mathcal{D})$, where $\mathbf{C}_{\mathcal{D}}^{-1} \in \mathbb{R}^{\Omega_{\mathcal{D}} \times \Omega_{\mathcal{D}}}$ is the diagonal matrix with values are $\frac{1}{k_j}$, for every datapoint $j$, for every item $x \in C_j$, and where $\Omega_{\mathcal{D}} = \sum_{i=1}^{m} k_i$. $X(\mathcal{D})$ should therefore be thought of as a scaled design matrix. Simple calculations show that,

$$L = \frac{1}{m} X(\mathcal{D})^T X(\mathcal{D}) \succeq 0.$$

As a consequence of the properties of matrix rank, we then have that rank$(L) = $ rank$(X(\mathcal{D})) = $ rank$(G(\mathcal{D}))$. Thus, from Theorem 4 of [48], we have that if the dataset $\mathcal{D}$ identifies the CDM, rank$(L) = d - 1$, and hence $\lambda_2(L) > 0$. We may then leverage conditions for identifiability from [48] (Theorem 1 and 2) to determine when $L$ is positive definite.

With this matrix, we can write,

$$z^T \nabla^2 \ell(u) z \geq \beta_{k_{\max}} z^T L z = \beta_{k_{\max}} ||z||_L^2,$$

which is equivalent to stating that $\ell(u)$ is $\beta_{k_{\max}}$-strongly convex with respect to the $L$ semi-norm at all $u \in \mathcal{U}_B$. Since $u^\star, \hat{u}_{\text{MLE}} \in \mathcal{U}_B$, we can now follow the implications of strong convexity with respect to the $L$ semi-norm used in Theorem 1, of course now with a different $L$, and conclude that:

$$||\hat{u}_{\text{MLE}} - u^\star||_L^2 \leq \frac{1}{\beta_{k_{\max}}^2} \nabla \ell(u^\star)^T L^\dagger \nabla \ell(u^\star).$$

Now, all that remains is bounding the term on the right hand side. Recall the expression for the gradient

$$\nabla \ell(u^\star) = -\frac{1}{m} \sum_{j=1}^{m} \sum_{i=1}^{k_j} \mathbf{1}[y_j = i] E_{j,k_j} R_{i,k_j} \nabla \log(F_{k_j}(u^{\star T} E_{j,k_j} R_{i,k_j})) = -\frac{1}{m} \sum_{j=1}^{m} E_{j,k_j} V_{j,k_j},$$

where in the equality we have defined $V_{j,k_j} \in \mathbb{R}^{k_j}$ as

$$V_{j,k_j} := \sum_{i=1}^{k_j} \mathbf{1}[y_j = i] R_{i,k_j} \nabla \log(F_{k_j}(u^{\star T} E_{j,k_j} R_{i,k_j})).$$

It is not difficult to see that $V_{j,k_j}$ is defined identically to the one in Theorem 1. We may thus borrow three results directly from that proof:

$$V_{j,k_j}^T \mathbf{1} = 0 \qquad\qquad \mathbb{E}[V_{j,k_j}] = 0 \qquad\qquad \sup_{j \in [m]} ||V_{j,k_j}||_2^2 \leq 2.$$

The remainder of the proof departs from the corresponding sections of 1, and we thus carefully detail every step. Consider the matrix $M_k = (I - \frac{1}{k}\mathbf{1}\mathbf{1}^T)$. We note that $M_k$ has rank $k - 1$, with its nullspace corresponding to the span of the ones vector. We state the following identities:

$$M_k = M_k^\dagger = M_k^{\frac{1}{2}} = M_k^{\dagger \frac{1}{2}}.$$

Thus we have $M_{k_j} V_{j,k_j} = M_{k_j}{}^{\frac{1}{2}} M_{k_j}^{\frac{1}{2}} V_{j,k_j} = M_k M_k^\dagger V_{j,k_j} = V_{j,k_j}$, where the last equality follows since $V_{j,k_j}$ is orthogonal to the nullspace of $M_{k_j}$.

Next, we have

$$\mathbb{E}[\nabla \ell(u^\star)^T L^\dagger \nabla \ell(u^\star)] = \frac{1}{m^2} \mathbb{E}\Big[ \sum_{j=1}^m \sum_{l=1}^m V_{j,k_j}^T E_{j,k_j}^T L^\dagger E_{l,k_l} V_{l,k_l} \Big]$$

$$= \frac{1}{m^2} \mathbb{E}\Big[ \sum_{j=1}^m \sum_{l=1}^m V_{j,k_j}^T M_{k_j}{}^{\frac{1}{2}} E_{j,k_j}^T L^\dagger E_{l,k_l} M_{k_l}{}^{\frac{1}{2}} V_{l,k_l} \Big]$$

$$= \frac{1}{m^2} \mathbb{E}\Big[ \sum_{j=1}^m V_{j,k_j}^T M_{k_j}{}^{\frac{1}{2}} E_{j,k_j}^T L^\dagger E_{j,k_j} M_{k_j}{}^{\frac{1}{2}} V_{j,k_j} \Big]$$

$$\leq \frac{1}{m} \mathbb{E}\Big[ \sup_{l \in [m]} ||V_{l,k_l}||_2^2 \Big] \frac{1}{m} \sum_{j=1}^m \mathbf{tr}\Big( M_{k_j}{}^{\frac{1}{2}} E_{j,k_j}^T L^\dagger E_{j,k_j} M_{k_j}{}^{\frac{1}{2}} \Big)$$

$$= \frac{1}{m} \mathbb{E}\Big[ \sup_{l \in [m]} ||V_{l,k_l}||_2^2 \Big] \frac{1}{m} \sum_{j=1}^m \mathbf{tr}\Big( L^\dagger E_{j,k_j} M_{k_j}{}^{\frac{1}{2}} M_{k_j}{}^{\frac{1}{2}} E_{j,k_j}^T \Big)$$

$$= \frac{1}{m} \mathbb{E}\Big[ \sup_{l \in [m]} ||V_{l,k_l}||_2^2 \Big] \mathbf{tr}\Big( L^\dagger L \Big)$$

$$= \frac{1}{m} \mathbb{E}\Big[ \sup_{l \in [m]} ||V_{l,k_l}||_2^2 \Big] (d - 1),$$

where the second line follows from identities of the $M$ matrix, the third from the independence of the $V_{j,k_j}$, the fourth from an upper bound of the quadratic form, the fifth from the properties of trace, the sixth from the definition of the matrix $L$, and the last from the value of the trace, which is simply the identity matrix with one zero entry in the diagonal.

Bringing the final expression back to $\mathbb{E}[\nabla \ell(u^\star)^T L^\dagger \nabla \ell(u^\star)]$, we have that

$$\mathbb{E}[\nabla \ell(u^\star)^T L^\dagger \nabla \ell(u^\star)] \leq \frac{2(d-1)}{m}.$$

This inequality immediately yields a bound on the expected risk in the $L$ semi-norm, which is,

$$\mathbb{E}[||\hat{u}_{\text{MLE}} - u^\star||_L^2] \leq \frac{2(d-1)}{m\beta_{k_{\max}}^2}.$$

By noting that $||\hat{u}_{\text{MLE}} - u^\star||_L^2 = (\hat{u}_{\text{MLE}} - u^\star) L(\hat{u}_{\text{MLE}} - u^\star) \geq \lambda_2(L)||\hat{u}_{\text{MLE}} - u^\star||_L^2$, since $\hat{u}_{\text{MLE}} - u^\star \perp \text{null}(L)$, we can translate our above result into the $\ell_2$ norm:

$$\mathbb{E}[||\hat{u}_{\text{MLE}} - u^\star||_2^2] \leq \frac{2(d-1)}{m\lambda_2(L)\beta_{k_{\max}}^2}.$$

Now, setting

$$c'_{B,k_{\max}} := \frac{2}{\beta_{k_{\max}}^2} = 2\exp(4B)k_{\max}^2,$$

we retrieve the expected risk bound in the theorem statement,

$$\mathbb{E}\big[\,\|\hat{u}_{\text{MLE}}(\mathcal{D}) - u^\star\|_2^2\,\big] \leq c_{B,k_{\max}} \frac{n(n-1)}{m\lambda_2(L)}.$$

We close the expected risk portion of this proof with some remarks about $c_{B,k_{\max}}$. The quantity $\beta_{k_{\max}}$, defined in equation (23), serves as the important term that approaches 0 as a function of $B$ and $k_{\max}$, requiring that the former be bounded. Finally, $\lambda_2(L)$ is a parallel to the requirements on the algebraic connectivity of the comparison graph in [49] for the pairwise setting. Though the object $L$ here appears similar to the graph Laplacian $L$ in that work, there are major differences that are most worthy of further study.

**From expected risk to tail bound.** Our proof of the tail bound is a continuation of the expected risk bound proof. While the expected risk bound closely followed the expected risk proof of [48] and [49], our tail bound proof contains significant novel machinery. Our presentation seem somewhat circular, given that the tail bound itself integrates out to an expected risk bound with the same parametric rates (albeit worse constants), but we felt that to first state the expected risk bound was clearer, given that it arises as a stepping stone to the tail bound.

Recall again the expression for the gradient in Equation (24). Useful in our analysis will be an alternate expression:

$$\nabla \ell(u^\star) = -\frac{1}{m}\sum_{j=1}^{m} E_{j,k_j} V_{j,k_j} = -\frac{1}{m} E^T V,$$

where we have defined $V \in \mathbb{R}^{\Omega_{\mathcal{D}}}$ as the concatenation of all $V_{j,k_j}$, and $E \in \mathbb{R}^{\Omega_{\mathcal{D}} \times n}$, the vertical concatenation of all the $E_{j,k_j}$. Recall that $\Omega_{\mathcal{D}} = \sum_{i=1}^{m} k_i$.

For the expected risk bound, we showed that $V_{j,k_j}$ have expectation zero, are independent, and $\|V_{j,k_j}\|_2^2 \leq 2$. Considering again the matrix $M_k$, re call that we have $M_{k_j} V_{j,k_j} = M_{k_j}^{\frac{1}{2}} M_{k_j}^{\frac{1}{2}} V_{j,k_j} = M_k M_k^\dagger V_{j,k_j} = V_{j,k_j}$, where the last equality follows since $V_{j,k_j}$ is orthogonal to the nullspace of $M_{k_j}$. We may now again revisit the expression for the gradient:

$$\nabla \ell(\theta^\star) = -\frac{1}{m}\sum_{j=1}^{m} E_{j,k_j} V_{j,k_j} = -\frac{1}{m}\sum_{j=1}^{m} E_{j,k_j} M_{k_j}^{1/2} V_{j,k_j} := -\frac{1}{m} X(\mathcal{D})^T V,$$

where we have defined $X(\mathcal{D}) \in \mathbb{R}^{\Omega_{\mathcal{D}} \times n}$ as the vertical concatenation of all the $E_{j,k_j} M_{k_j}^{1/2}$, and the scaled design matrix described before.

Now, consider that

$$\nabla \ell(u^\star)^T \hat{L}^\dagger \nabla \ell(u^\star) = \frac{1}{m^2} V^T X(\mathcal{D}) \hat{L}^\dagger X(\mathcal{D})^T V.$$

We apply Lemma 3, a modified Hanson-Wright-type tail bound for random quadratic forms. This lemma follows from simpler technologies (largely Hoeffding's inequality) given that the random variables are bounded while also carefully handling the block structure of the problem.

In the language of Lemma 3 we have $V_{j,k_j}$ playing the role of $x^{(j)}$ and $\Sigma_{\mathcal{D}} := \frac{1}{m^2} X(\mathcal{D}) \hat{L}^\dagger X(\mathcal{D})^T$ plays the role of $A$. The invocation of this lemma is possible because $V_{j,k_j}$ is mean zero, $\|V_{j,k_j}\|_2 \leq \sqrt{2}$, and because $\Sigma_{\mathcal{D}}$ is positive semi-definite. We sweep $K^4 = 4$ from the lemma statement into the constant $c$ of the right hand side. Stating the result of Lemma 3 we have, for all $t > 0$,

$$\mathbb{P}(V^T \Sigma_{\mathcal{D}} V - \sum_{i=1}^{m} \lambda_{\max}(\Sigma_{\mathcal{D}}^{(i,i)}) \mathbb{E}[V^{(i)T} V^{(i)}] \geq t) \leq 2\exp\left(-c\frac{t^2}{\sum_{i,j}\sigma_{\max}(\Sigma_{\mathcal{D}}^{(i,j)2})}\right). \qquad (25)$$

We note that

$$\sum_{i,j}\sigma_{\max}(\Sigma_{\mathcal{D}}^{(i,j)})^2 \leq \sum_{i,j}\sum_{k}\sigma_k(\Sigma_{\mathcal{D}}^{(i,j)})^2 = \sum_{i,j}\|\Sigma_{\mathcal{D}}^{(i,j)}\|_F^2 = \|\Sigma_{\mathcal{D}}\|_F^2,$$

where the first inequality follows because the max is less than the sum of positive values, and the first equality from the definition of Frobenius norm, and second from the Frobenius norm of blocks being

the Frobenius norm of the whole. This inequality allows us to conclude that, for the right hand side of of Equation (25):

$$2 \exp\left(-c \frac{t^2}{\sum_{i,j} \sigma_{\max}(\Sigma_{\mathcal{D}}^{(i,j)})^2}\right) \leq 2 \exp\left(-c \frac{t^2}{||\Sigma_{\mathcal{D}}||_F^2}\right).$$

Next, within the left hand side of Equation (25) we have,

$$\sum_{i=1}^m \lambda_{\max}(\Sigma_{\mathcal{D}}^{(i,i)}) \leq \sum_{i=1}^m \mathrm{tr}(\Sigma_{\mathcal{D}}^{(i,i)}) \leq \mathrm{tr}(\Sigma_{\mathcal{D}}),$$

and so:

$$\mathbb{P}(V^T \Sigma_{\mathcal{D}} V - \mathrm{tr}(\Sigma_{\mathcal{D}}) \sup_{i \in [m]} \mathbb{E}[V^{(i)T} V^{(i)}] \geq t) \leq \mathbb{P}(V^T \Sigma_{\mathcal{D}} V - \sum_{i=1}^m \lambda_{\max}(\Sigma_{\mathcal{D}}^{(i,i)}) \mathbb{E}[V^{(i)T} V^{(i)}] \geq t).$$

We may now combine these improvements for a much more compact version of Equation (25):

$$\mathbb{P}(V^T \Sigma_{\mathcal{D}} V - \mathrm{tr}(\Sigma_{\mathcal{D}}) \sup_{i \in [m]} \mathbb{E}[V^{(i)T} V^{(i)}] \geq t) \leq 2 \exp\left(-c \frac{t^2}{||\Sigma_{\mathcal{D}}||_F^2}\right). \tag{26}$$

Now, some algebra (see Lemma 2) reveals that:

$$\mathbf{tr}(\Sigma_{\mathcal{D}}) = \frac{d-1}{m}, \qquad\qquad ||\Sigma_{\mathcal{D}}||_F^2 = \frac{(d-1)}{m^2}, \tag{27}$$

where we have used the fact that $\hat{L} = \frac{1}{m} X(\mathcal{D})^T X(\mathcal{D})$ and hence its pseudoinverse cancels out the other terms in $\Sigma_{\mathcal{D}}$. Now, noting that the norm of $V_{i,k_i}$ is bounded (thus $\mathbb{E}[V^{(i)T} V^{(i)}] \leq 2$), and substituting in the relevant values into Equation (26), we have for all $t > 0$:

$$\mathbb{P}\left(\nabla \ell(u^\star)^T \hat{L}^\dagger \nabla \ell(u^\star) - \frac{2(d-1)}{m} \geq t\right) \leq 2 \exp\left(-c \frac{m^2 t^2}{d-1}\right).$$

A variable substitution and simple algebra transforms this expression to

$$\mathbb{P}\left[\nabla \ell(u^\star)^T \hat{L}^\dagger \nabla \ell(u^\star) \geq c_2 \frac{t(d-1)}{m}\right] \leq e^{-t} \quad \text{for all } t > 1,$$

where $c_2$ is an absolute constant. We may then make the same substitutions as before with expected risk, to obtain,

$$\mathbb{P}\left[||\hat{u}_{\mathrm{MLE}}(\mathcal{D}) - u^\star||_2^2 > c_2 \frac{t(d-1)}{m \lambda_2(L) \beta_{k_{\max}}^2}\right] \leq e^{-t} \quad \text{for all } t > 1.$$

Setting $d - 1 = n(n-1) - 1$, dropping the final minus one term, and making the appropriate substitution with $c_{B,k_{\max}}$, we retrieve the desired tail bound statement, for another absolute constant $c$.

$$\mathbb{P}\left[||\hat{u}_{\mathrm{MLE}}(\mathcal{D}) - u^\star||_2^2 \geq c_{B,k_{\max}} \frac{tn(n-1)}{m \lambda_2(L)}\right] \leq e^{-t} \quad \text{for all } t > 1.$$

Integrating the above tail bound leads to a similar bound on the expected risk (same parametric rates), albeit with a less sharp constants due to the added presence of $c$. $\qquad\square$

## Footnotes

[2]Preflib data is available at: http://www.preflib.org/

[3] A semi-norm is a norm that allows non-zero vectors to have zero norm.