[Reviews · NeurIPS 2020]

Review 1

Summary and Contributions: This work proposes a novel modeling of ranking, called "contextual repeated selection (CRS)", that provides natural multimodality to the ranking space. Theoretical guarantees for the performance under the CRS model is provided, and experiments are conducted on a variety of ranking domains.

Strengths: * Connecting ranking with choice to create choice-based model of ranking data is the main contribution of this paper. The proposed contextual repeated selection subsumes well known ranking models by applying repeated selection to choice models. This model does not assume transitivity or center the distribution around aa single total ordering, and it can be efficiently learned as shown in the experiment results. * Rigorous theoretical guarantee are provided for the maximum likelihood estimation for MNL and PL, and convergence guarantees for the CRS model are demonstrated. * Empirical results show that the proposed model offers better performance for a variety of ranking domains.

Weaknesses: Generally speaking, the paper is interesting with solid theoretical results. I have few questions: * In the analysis of the CRS model, the full CRS is considered. However, the factorized CRS is considered for the empirical results. Do the derived theoretical guarantees follow for the factorized CRS? Is it possible to simulate both versions of CRS to visualize the performance gap, if any? * In the empirical results section, It is evident that the performance gap between PL and CRS is much lower than the one between CRS and Mallows across most of the databases. Are there any insights about such a behavior?

Correctness: For the main paper, the reviewer believes that the theorems sound correct. However, the reviewer did not thoroughly proofread the supplementary materials.

Clarity: The paper is generally well written. However, there are many typos in the main paper that should be corrected. Examples are "that is crucial our paper's framework", "our new model perform", "Repeated these choices until", etc.

Relation to Prior Work: Yes, the literature review is extensive, and the contribution of this paper is discussed and compared against previous contributions.

Reproducibility: Yes

Additional Feedback: ################## Update after Rebuttal ################## I would like to thank the authors for addressing all the major comments raised by the reviewers in their feedback. Hence, I will keep my score the same.


Review 2

Summary and Contributions: The paper proposes a repeated selection interpretation of the recently introduced Context dependent utlity model, a discrete choice model, for defining potentially non-unimodal distributions over permutations. The theoretical guarantees of the maximum likelihood estimator for such a model is analyzed.

Strengths: - Extends the discrete choice model to a novel model for permutations (Models of intransitivity is not as well understood as the transitive models) - Theoretically sound and the techniques end up improving guarantees known for existing models such as Plackett- Luce and MNL

Weaknesses: - As the authors point out, the analysis seems to be sub optimal when compared to what is observed in practice. This is expected because there is no clean way to recover the previous models based on further assumptions about the parameters to be learnt. - More specifically, it would have been nice to see a dependency on the rank of the factor matrices C,T showing up in the tail bounds. The rank determines the degree of freedom of the parameters and hence one would expect the learning rates to depend on these. Also, how does the rank play a role in identifiability? The authors mention one needs O(nlogn2) samples for the model to be identifiable with high probability. Is this independent of the rank?

Correctness: - The experimental results seem to show that increasing the rank typically increase the average out of sample negative log likelihood. It would have been nice to see a study on the effect on synthetic data where the underlying rank of the parameter factors known.

Clarity: - A reasonably well written paper

Relation to Prior Work: The work depends on and extends the recently introduced CDM for discrete choice. The authors do a reasonable job of reviewing the previous contributions.

Reproducibility: Yes

Additional Feedback:


Review 3

Summary and Contributions: The authors extend the recently proposed CDM rich choice model to rankings via contextual repeated selection (CRS). The main results of the paper are bounds on the expected risk as well as risk tail bounds, ie, guarantees on the speed of convergence of the ML estimate. As special cases, they also provide bounds and the expected risk of ML for MNL and PL models. Some limited experiments with real data show that CRS captures additional structure that in particular plain Plackett-Luce misses.

Strengths: - This is a well written paper. The concepts are clearly discussed and explained. (but see Weaknesses regarding utilization of the page budget.) - A ranking model that permits richer interaction between items than in classical PL et al is interesting, even though the generalization of CDM to CRS is relatively straightforward. - The main result of the paper, Theorem 3, which finds the theoretical bound for the risk of ML estimate is nice, as are the Theorems 1 and 2 which as a byproduct analyze MNL and PL models. Their bounds strengthen those from the literature for the PL model in particular.

Weaknesses: - While the theoretical bounds are nice, no algorithmic results for efficiently estimating the CRS model is given. I suppose that this all relies directly on the machinery developed for CDM in [48]. - I also find the utilization of the page limit very suboptimal. Main discussions such as those about the discrepancy of the optimization should not appear in the Appendix. - One important limitation is that the theorems provide guarantees only for datasets of *full* rankings. In many practical scenarios, the data would consist of partial rankings over (many small) subsets of items. Similar guarantees for this case would be very helpful, and would strengthen the contribution. - Given the parameter scaling of the model (quadratic in the number of items) it might not be easily applicable in practice. In fact I find the simulation results confusing. It is not clear why CRS has such nonlinear performance. I do not find the explanation on line 522-527 satisfactory.

Correctness: The proofs are quite long. From my quick scanning of the derivations, they look correct.

Clarity: Yes the paper is well written but I believe the main body should be shortened and more space should be dedicated to empirical results.

Relation to Prior Work: This is very thorough and well written.

Reproducibility: Yes

Additional Feedback: - the text cites some equations by #, but equations are not actually labeled - line 102: "...crucial *to* our paper" - line 160: should be \sigma_0


Review 4

Summary and Contributions: This paper proposes a CRS (contextual repeated selection) model, which is a generalization of Plackett-Luce model. The authors bound the risk (an indicator of MLE) for MNL model, Plackett-Luce model, and the proposed CRS model. Experiments show better fitness of CRS model to real-world data.

Strengths: 1. A novel context-dependent ranking model that is shown to better fit the real-world data; 2. Theoretical bounds on the performance of MLE for MNL, PL, and finally CRS with rigorous proofs.

Weaknesses: 1. It is hard to find the formal definition of the proposed CRS model. It seems to be the equation after line 175, but the authors did not say it explicitly. 2. Theorem 1 does not seem to be very challenging as it has \lambda_2(L) in the bounds. It would be more interesting if \lambda_2(L) can be bounded. 3. In the tail bound of Theorem 2 (equation between lines 259 and 260), it seems this bound becomes trivial for very large n since the last term can be greater than one. 4. The authors' understanding of identifiability is not correct. Identifiability is a property of a model assuming infinite data can be observed. You can not say a model is not identifiable due to insufficient data.

Correctness: I did not see obvious errors. Theorems 1 and 2 are not very surprising to me, but okay as intermediate steps to build Theorem 3.

Clarity: Yes.

Relation to Prior Work: Yes.

Reproducibility: Yes

Additional Feedback: About the "identifiability" discussed in this paper. I suggest using the conditions on data as conditions for MLE to converge. In order to characterize identifiability, the authors have to define different models and prove that some models are identifiable while others are not. I have read all reviews and the authors' response. My score is updated.

[Author Response · NeurIPS 2020]

We thank all the reviewers for their time and feedback.

**R1.** We enumerate to match R1's questions:
**1**. Every factorized CRS model class is a subset of the full CRS model class, and hence our upper bounds on the error of the full-CRS extend to the factorized-CRS. It is likely that low-rank models admit an improved rank-dependent convergence rate, and we agree that a simulation of the presumed gap would be a valuable addition. We plan to revise this.
**2**. Inferring the Mallows Model's parameters is NP-hard, and the Mallows Greedy Approximation (MGA) only provides an approximate solution. We believe at least part of the larger performance gap between CRS and Mallows (compared to CRS and PL) can be attributed to the shortcomings of the approximate solution.

**R2.** Regarding the role of rank in the learning rate of factorized CRS models: We agree this is a good point, also raised by R1; see our answer to their Q1. Regarding rank and identifiability, the factorized model is identifiable if the full model is. In Ref [48] on the CDM choice model, their Thm 4, they give a linear-algebraic sufficient condition for identifiability of the full CDM, in terms of the sets compared. It's possible that the condition could be turned into an improved rank-dependent sufficient condition for low-rank factorized models; we haven't tried to refine that aspect of their work. Regarding the necessity of $O(n \log(n)^2)$ samples: that is for the full CRS. A possible conjecture for the rank-$r$ factorized case would be $O(r \log(n)^2)$, but we haven't tried to prove that.

**R3.** We enumerate to match R3's questions and comments:
**1**. For algorithmic results, we indeed rely on the machinery developed for CDM in [48]. More specifically, we rely on the well-established and general results that smooth and differentiable strongly convex functions in a bounded parameter space are guaranteed to efficiently converge using first order gradient methods. These methods are the ones we also use in the empirics section.
**2**. Good suggestion, we should include the discrepancies in optimization in the main text.
**3**. We strongly agree guarantees for partial rankings would strengthen the contribution, and found it difficult to resist the temptation to include such results. Due to space constraints, we ultimately prioritized clarity in development and exposition, and felt the results would find a home in a lengthier journal version of the contribution, also covering partial rankings and top-$k$ rankings.
**4**. The quadratic parameter scaling is only for the full CRS model; the factorized model, used in our empirical work, scales linearly (at a fixed rank). Regarding Figure 2: it is confusing, we apologize. The point of the figure is to show that CRS is doing particularly well, relative PL, at predicting second and third position choices. We will re-consider the logL-based position measure, perhaps using probabilities or accuracy instead.

**R4.** We enumerate to match R4's questions and comments:
**1**. The CRS model is defined after line 175; we will try to make this definition clearer during revisions.
**2**. Regarding Thm 1, our use of $\lambda_2(L)$ in the result of Theorem 1 is deliberate, to characterize the intricate choice set structure-dependence on the rate of error convergence. It is also consistent with the literature; see [49] for examples of how $\lambda_2(L)$ can vary considerably with different choice set structures. From [49], $\lambda_2(L)$ also appears in the lower bound of the error, making it a fundamental quantity to the rate of error convergence. Our tight upper bound in Thm 1 improves upon multiple prior attempts [24, 49].
**3**. Regarding Thm 2, our tail bound is not trivial for large $n$; its claim is that so long as the number of samples $\ell$ grows linearly with the number of parameters (items) $n$, the error obeys an exponential tail. That is, it is bounded with exponentially high probability. Our result is in-fact tight, as demonstrated by a lower bound in [24]: no algorithm to infer PL parameters can do so without requiring $\ell$ on the order of $n$.
**4**. Thank you for raising this point. Our use of "identifiability" stems from discrete choice settings, where choices conditioned on choice sets are treated as the data, but choice sets define the support of the (model) distribution. Without conditions on the choice set structure (e.g., encoded by $\lambda_2(L)$) a choice model cannot be identified even with infinite data. Thought of in this manner, we believe identifiability is the correct term.
   For repeated selection models of rankings like PL and CRS, however, the choice set distribution is also fixed by the ranking model parameters, and for the class of model parameters we consider, both PL and CRS are always identifiable. The claims we make about CRS and PL needing a particular number of rankings should be amended to be claims about *estimability*—the number of rankings before the parameters are no longer under-determined—and *not identifiability*. We thus apologize for our misuse of the word identifiability and the resulting confusion. We will amend it.



[Meta-Review · NeurIPS 2020]

The paper proposes a new "CRS" model for ranking, provides theoretical guarantees for MLE (which also apply to MNL as a special case), and real-world experiments. Given the very nice conceptual, theoretical and empirical results, this paper is recommended for acceptance. The paper has the wrong style file for the rebuttal. While the rebuttal is not ignored for this reason since the rebuttal at least obeys the spirit of the style file, request the authors to be much more careful in the future.